# SARS-CoV-2 viral persistence in lung alveolar macrophages is controlled by IFN-γ and NK cells

Nicolas Huot [1] ✉, Cyril Planchais[2], Pierre Rosenbaum[2], Vanessa Contreras[3], Beatrice Jacquelin [1], Caroline Petitdemange[1], Marie Lazzerini[1], Emma Beaumont[1], Aurelio Orta-Resendiz[1], Félix A. Rey [4], R. Keith Reeves [5,6,7,8], Roger Le Grand [3], Hugo Mouquet[2] & Michaela Müller-Trutwin[1]

Severe acute respiratory syndrome coronavirus 2 (SARS-CoV-2) RNA generally becomes undetectable in upper airways after a few days or weeks postinfection. Here we used a model of viral infection in macaques to address whether SARS-CoV-2 persists in the body and which mechanisms regulate its persistence. Replication-competent virus was detected in bronchioalveolar lavage (BAL) macrophages beyond 6 months postinfection. Viral propagation in BAL macrophages occurred from cell to cell and was inhibited by interferon-γ (IFN-γ). IFN-γ production was strongest in BAL NKG2r⁺CD8⁺ T cells and NKG2A^lo natural killer (NK) cells and was further increased in NKG2A^lo NK cells after spike protein stimulation. However, IFN-γ production was impaired in NK cells from macaques with persisting virus. Moreover, IFN-γ also enhanced the expression of major histocompatibility complex (MHC)-E on BAL macrophages, possibly inhibiting NK cell-mediated killing. Macaques with less persisting virus mounted adaptive NK cells that escaped the MHC-E-dependent inhibition. Our findings reveal an interplay between NK cells and macrophages that regulated SARS-CoV-2 persistence in macrophages and was mediated by IFN-γ.

Severe acute respiratory syndrome coronavirus 2 (SARS-CoV-2) replicates in both upper and lower airways, with viral RNA detectable days before symptoms, peaking within the first week, and gradually declining due to host antiviral responses[1]. However, cellular and humoral immune responses, as well as the activation of neural pathways, can collectively contribute to distant inflammatory effects[2]. Furthermore, in some SARS-CoV-2-infected individuals, complete clearance of the virus does not appear to occur over extended periods[3,4].

Macrophages (Mac) and natural killer (NK) cells are frontline innate effector cells against pathogens. Mac responds to microbial threats by producing inflammatory molecules, phagocytosing pathogens and promoting tissue repair. Dysregulated Macs, as

[1]Institut Pasteur, Université Paris-Cité, HIV, Inflammation and Persistence Unit, Paris, France. [2]Institut Pasteur, Université Paris Cité, INSERM U1222, Humoral Immunology Unit, Paris, France. [3]Université Paris-Saclay, INSERM, CEA, Immunologie des Maladies Virales, Auto-Immunes, Hématologiques et Bactériennes (IMVA-HB/IDMIT/UMR1184), Fontenay-aux-Roses & Kremlin Bicêtre, France. [4]Institut Pasteur, Université Paris-Cité, Structural Virology Unit, CNRS UMR3569, Paris, France. [5]Center for Virology and Vaccine Research, Beth Israel Deaconess Medical Center, Harvard Medical School, Boston, MA, USA. [6]Division of Innate and Comparative Immunology, Center for Human Systems Immunology, Department of Surgery, Duke University School of Medicine, Durham, NC, USA. [7]Ragon Institute of Massachusetts General Hospital, MIT, Cambridge, MA, USA. [8]Duke Research and Discovery at RTP, Duke University Health System, Durham, NC, USA. ✉e-mail: nicolas.huot@pasteur.fr

observed during SARS-CoV-2 infection, can harm the host, such as in the infection-induced Mac activation syndrome[5]. Mac comprises about 70% of the total leukocyte population in the lung[6] and can be exploited by various viruses, including respiratory syncytial virus, for dissemination, long-term tissue persistence and virus replication[7]. SARS-CoV-2 can infect monocytes and Macs through different pathways, but whether the virus can complete its lifecycle in these cells is still debated[8,9].

NK cells are crucial innate immune responders responsible for viral clearance and can modulate adaptive T and B cell responses[10,11]. However, their contribution to SARS-CoV-2 immunity remains unclear[12,13]. High NK cell levels in blood correlate with a rapid viral load decline, but NK cell frequency often decreases in the blood during early SARS-CoV-2 infection, and NK cells display an exhaustion phenotype[14]. While NK cells from healthy individuals can directly kill SARS-CoV-2-infected cells in vitro, those from moderate or severe cases exhibit impaired cytotoxic activity[15–17].

Nonhuman primate (NHP) models are vital tools for understanding immune cell functioning and responses in tissues, particularly when studying human infectious diseases[18]. We used an NHP model (cynomolgus macaques) to investigate lung-resident Mac and NK cell responses to SARS-CoV-2 (wild-type or Omicron strains) infection. We detected replication-competent SARS-CoV-2 in bronchoalveolar fluid (BALF) Mac for up to 18 months after infection. Interferon-γ (IFN-γ) inhibited SARS-CoV-2 replication in these BALF Mac. However, IFN-γ production was reduced in NK cells from some macaques. IFN-γ facilitated the resistance to NK cell-mediated killing through upregulation of major histocompatibility complex (MHC)-E on the cell surface of bronchioalveolar lavage (BAL) Mac, which promoted the persistence of the infected cells.

## Results

### SARS-CoV-2 induces long-term alterations in Mac phenotype

To explore long-term SARS-CoV-2 effects on innate immunity, we infected 25 cynomolgus macaques with wild-type (hereafter WTM, $n = 15$), Omicron BA.1 ($n = 6$) and Omicron BA.2 ($n = 4$) variants (hereafter OM), along with six noninfected macaques as controls (HC; Fig. 1a and Extended Data Table 1). Viral RNA loads peaked in nasal and tracheal swabs at day 3 postinfection (p.i.) in WTM and OM ($7.9 \times 10^8$ and $2.78 \times 10^7$ copies per ml, respectively; Fig. 1a). Viral RNA was higher in the nasal swabs from WTM than OM (Extended Data Fig. 1a). By day 21, all macaques tested negative for SARS-CoV-2 RNA in nasal and tracheal swabs and remained negative by this readout up to 18 months p.i. (Fig. 1a). Immune responses were assessed at a median of 221 d p.i., with the analysis potentially extending to day 479 (Supplementary Table 1). Plasma immunoglobulin G (IgG) and IgA reactivities against spike and receptor-binding domain (RBD) were comparable in WTM and OM (Fig. 1b and Extended Data Fig. 1b). Inflammatory cytokines (interleukin (IL)-6, IL-18, IL-23, CXCL10) were higher in WTM and OM at 221 d.p.i. compared to HC (Extended Data Fig. 1c), suggesting lasting inflammation.

Multiparameter flow cytometry of BALF cells at 221 d p.i. (median) indicated higher frequency of CD45+CD64+ Mac (hereafter Mac) and lower lymphocyte frequencies of CD45+ lymphocyte in WTM than HC (Fig. 1c and Extended Data Fig. 1d). Mac and lymphocyte frequencies correlated with plasma CXCL10, IL-23 and IL-6 (Extended Data Fig. 1e), indicating inflammation-linked dysregulation. BALF CD45+CD64+ Mac from WTM and OM showed distinct clustering from HCs. They displayed higher expression of CD206, CD4, CD11c, MHC-E and IL-10 and lower expression of CD16 and CXCR4 compared to HC (Fig. 1d,e and Extended Data Fig. 1f,h). They also expressed MHC-E, CD11c and IL-10 more frequently, with fewer CD16+ Mac in WTM and OM compared to HC (Fig. 1f,g), suggesting an alternative activation of alveolar Mac, which may indicate lung contusion and pneumonia[19].

Quantification of cytokine production by BALF Macs 8 h post-lipopolysaccharide (LPS) stimulation indicated lower IFN-γ in WTM than HC (Fig. 1h and Extended Data Fig. 2). WTM BALF Mac also exhibited reduced viability after 8 h in culture with or without LPS, compared to HC (Fig. 1i,j). Mac viability correlated positively with IFN-γ production at 8 and 12 h post-LPS stimulation (Fig. 1k), suggesting altered activation in SARS-CoV-2-infected Mac, particularly in WTM. These findings indicated a lasting SARS-CoV-2-induced phenotype change in BALF Mac, marked by increased cell frequency, increased expression of MHC-E, IL-10 and CD11c and impaired IFN-γ production in response to LPS, compared to HC.

### Replication-competent SARS-CoV-2 is detected in BALF Mac

Given the persisting inflammation and phenotypic modifications of the BALF Mac, we examined the persistence of viral RNA in BALF cells using reverse transcription polymerase chain reaction (RT–PCR). SARS-CoV-2 RNA was detected in 12 of 15 WTM and 5 of 10 OM (Fig. 2a). Viral RNA levels inversely correlated with BALF Mac survival at 24 h (Fig. 2a). Based on viral RNA levels, macaques were grouped as high expressors (>1ΔΔCt that included 6 WTM (hereafter (WTM^hi) and 2 OM (hereafter OM^hi)), low expressors (<1ΔΔCt, 6 WTM^lo and 3 OM^lo) and no expression (3 WTM^neg and 5 OM^neg; Extended Data Fig. 2d). This clustering unveiled that WTM^hi and OM^hi showed higher IL-18, CXCL10 and sCD14 levels compared to HC (Extended Data Fig. 2e).

We next assessed if Mac were a viral source by quantifying spike protein in BALF CD45+CD64+ Mac using intracellular flow cytometry (Fig. 2b). Spike+ Mac was detected in 12 of 15 WCM and 8 of 10 OCM (Fig. 2c). Spike+ Mac frequency was 41% in WTM^hi and OM^hi, 25% in WTM^lo and OM^lo and 8.6% in WTM^neg and OM^neg (Fig. 2c). Spike+ Mac frequency correlated positively with viral RNA in total BALF cells and negatively with BALF Mac survival in culture (Fig. 2c).

To further assess SARS-CoV-2 replication in Mac, we examined double-stranded RNA and nonstructural protein 3 (NSP3) protein expression in BALF Mac over an 18-h culture period (Fig. 2d–f). At 8 h of culture, viral dsRNA and NSP3 proteins were increased in BALF Mac in all 8 WTM^hi or OM^hi, 5 of 9 WTM^lo or OM^lo and 3 of 8 WTM^neg or OM^neg (Fig. 2f and Extended Data Fig. 3a). Spike protein further increased over time in all WTM^hi/OM^hi and WTM^lo/OM^lo, but not in 4 of 8 OM^neg macaques, suggesting SARS-CoV-2 had the ability to replicate in Mac. We also noticed an increased number of BALF Mac syncytia during the culture period, likely mediated by spike (Extended Data Fig. 3b), and filiform extensions in BALF Mac at 8 h of culture (Extended Data Fig. 3b), which facilitated connections between Macs (Fig. 2g). Confocal images showed NSP3 protein within some of these protrusions (Fig. 2d and Extended Data Fig. 3b), suggesting viral replication in BALF Mac through cell-to-cell propagation.

To study the impact of viral replication in Macs, we analyzed the transcriptome of BALF CD45+CD64+ Mac from 5 randomly selected WTM, 10 OM and 6 HC. Mac was cultured for 8 h, to allow SARS-CoV-2 protein expression. We used a NanoString nCounter codeSet targeting 753 NHP genes involved in immune responses to estimate Mac transcriptomes during viral replication in culture (Fig. 3a). Principal component analysis (PCA) showed the separation of WCM, OCM and HC (Fig. 3b). WTM and OM displayed 138 differentially regulated genes compared to HC (Fig. 3c and Extended Data Fig. 4a,b). Gene set enrichment analysis (GSEA) showed one significantly enriched gene set (lipid_binding) and ten decreased gene sets (such as cytokine_acticity, chemokine_activity, signaling receptor_regulator_activity and cytokine receptor binding) in all WTM Macs and OM Macs compared to HC Macs (Fig. 3d and Extended Data Fig. 4c,d). Decreased gene sets in the Mac were associated with chemotaxis and cytokines (*CXCL1, CXCL6, IFNG* and *TGFB1*), while upregulated genes included *FN1, CD1b, S100A8* and *S100A9* (Fig. 3e and Extended Data Fig. 4a,b). These observations suggested that SARS-CoV-2 exhibited prolonged persistence and active replication within BALF Mac in infected macaques.

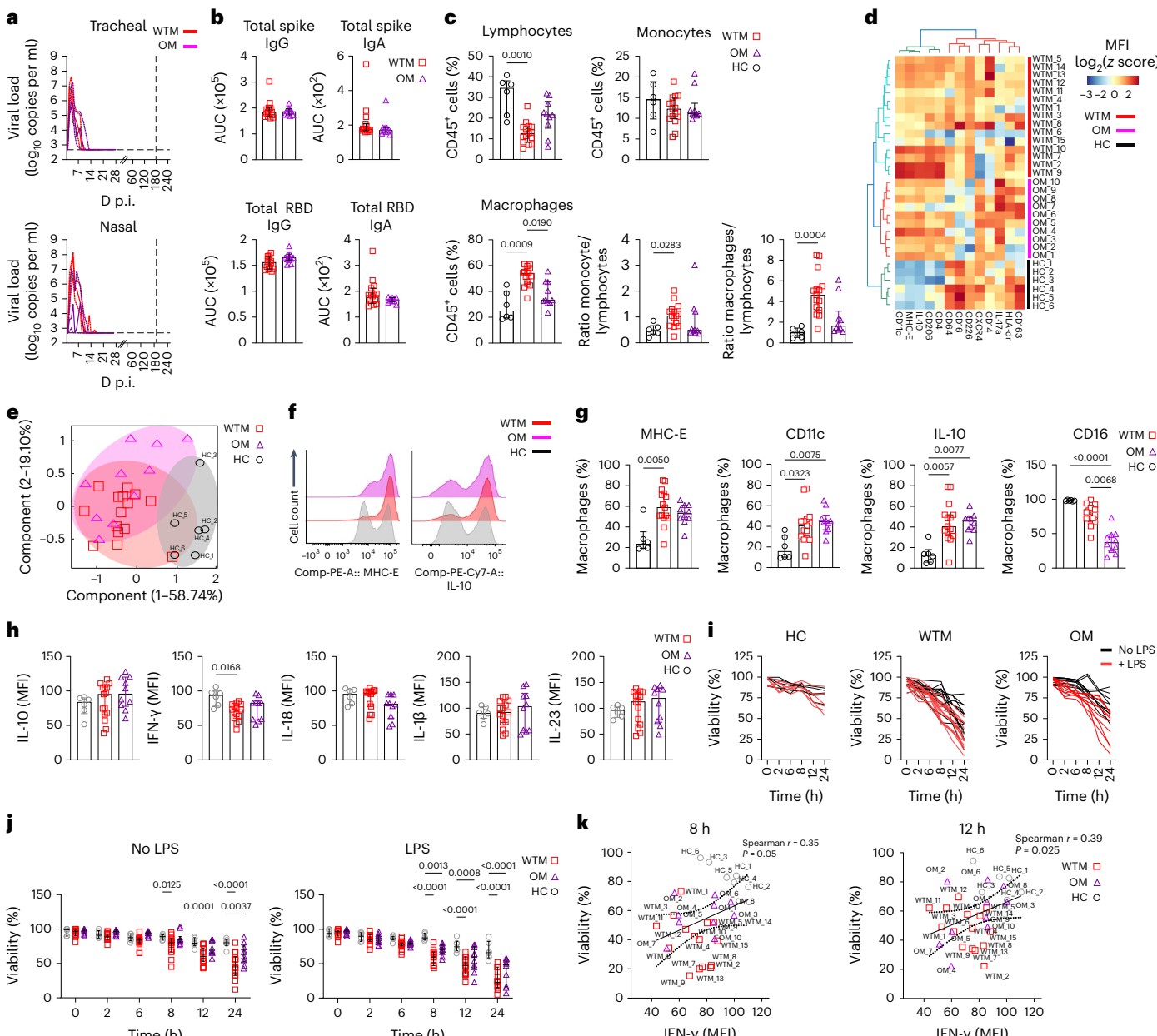

**Fig. 1 | Increased expression of MHC-E and IL-10 in BALF Mac from SARS-CoV-2-infected macaques. a**, PCR quantification of tracheal and nasal swab viral loads in 25 cynomolgus macaques infected with wild-type SARS-CoV-2 (WTM, *n* = 15) or Omicron variants (OM, *n* = 10) at a median of 221 d p.i. **b**, Plasma titers of spike and RBD IgG and IgA in WTM and OM at 221 d p.i. (median). **c**, Percentage and ratio of lymphocytes, monocytes and macrophages in BALF from HC, WTM and OM at a median of 221 d p.i. **d**, Hierarchical clustering heatmaps displaying log₂ *z* score expression of proteins in BALF Mac from HC, WTM and OM at a median at a median of 221 d p.i. **e**, PCA of protein expression in BALF Mac as in **c** depicting clustering patterns among macaques. **f**, Expression of MHC-E (top) and

IL-10 (bottom) in BALF Mac as in **c**. **g**, Frequency of MHC-E, CD11c, IL-10 and CD16 in BALF Mac as in **c**. **h**, Median expression per cell of cytokines in BALF Mac after 8-h LPS stimulation isolated from infected macaques as in **c**. **i,j**, Viability of BALF Mac isolated as in **c**, and cultured with and without LPS stimulation. **k**, Spearman correlation analysis between intracellular IFN-γ expression and viability at 8 and 12 h post-LPS exposure in BALF Macs as in **c**. Each symbol represents an individual macaque. In all graphs, bars indicate medians and interquartile ranges are displayed. Statistical tests: Mann–Whitney (**b**), Kruskal–Wallis with Dunn's post hoc test (**c**, **g** and **h**) and nonparametric Wilcoxon signed-rank test (**k**). Linear regression lines and confidence intervals are shown in correlation analyses.

## IFN-γ⁺ NK cells are reduced in WTM and OM BALF

To explore the potential involvement of immune mechanisms in constraining viral replication in the BALF Mac, we conducted an in-depth phenotypical analysis of lymphocytes in BALF. After using a 20-color marker flow panel on total BALF cells from WTM, OM and HC, unbiased clustering revealed nine distinct lymphocyte populations, subsequently validated through manual gating (Fig. 4a and Extended Data Fig. 4e). WTM and OM BALF had an increased frequency of Lin⁻NKG2R⁺ NK cells compared to HC (Extended Data Fig. 4f). Specifically, WTM had

higher counts of NKG2R⁺CD8⁺ T cells and CD3⁺CD4⁻CD8⁻ T cells. This higher frequency of Lin⁻NKG2R⁺ NK cells and NKG2R⁺CD8⁺ T cells correlated with elevated expression of NKG2R and EOMES in total BALF cells from WTM, compared to HC (Extended Data Fig. 4g). In contrast, OM showed an abundance of CD34⁺ cells (Extended Data Fig. 4f), potentially accounting for the heightened expression of CD34 and CD117 in total BALF cells compared to HC (Extended Data Fig. 4g). Total BALF lymphocytes in OM exhibited increased expression of IFN-γ compared to HC (Extended Data Fig. 4g). We observed no variations in the prevalence

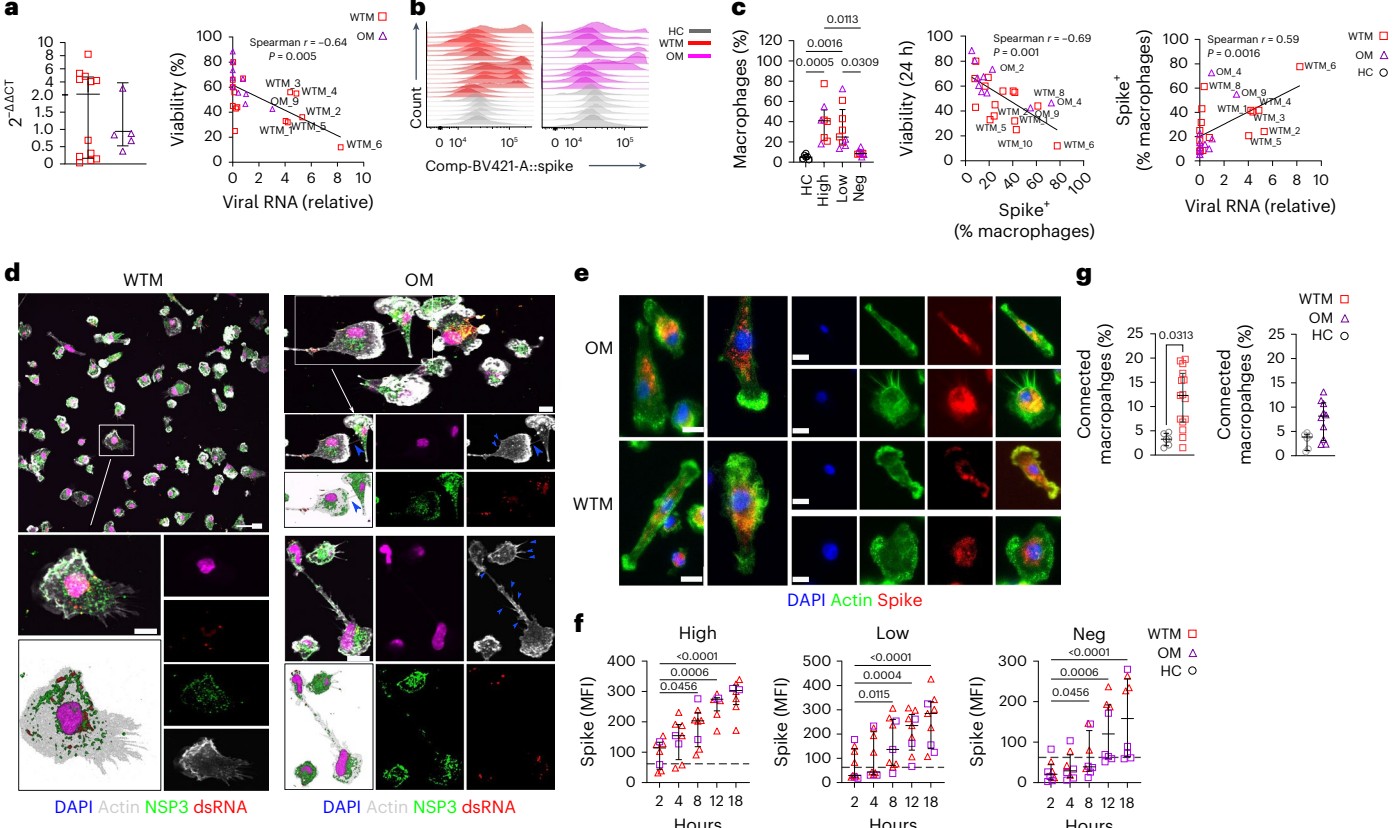

**Fig. 2 | SARS-CoV-2 is expressed in BALF Mac from WTM and OM at more than 221 d p.i. a**, PCR analysis of SARS-CoV-2 RNA in total BALF cells from WTM (*n* = 15) and OM (*n* = 10) at a median of 221 d p.i. (left) and correlation between 24-h macrophage viability (as in Fig. 1i) and viral RNA expression (right). **b**, Intracellular spike SARS-CoV-2 expression in BALF Mac from WTM and OM as in **a**, assessed by flow cytometry. **c**, Frequency of Spike⁺ BALF Mac from WTM and OM as in **a** (left) and correlation between Spike⁺ Mac frequency and viability at 24 h (right). Background for spike staining set at 5.28% based on HC. **d**, Confocal microscopy images of BALF Mac isolated from WTM (*n* = 15) and OM (*n* = 10) ≥ 221 d p.i., cultured for 8 h, and stained with DAPI, actin-phalloidin, dsRNA and NSP3 Abs. Blue arrowheads indicate supposed tunneling nanotubes

(TNT). Scale bars, 10 µm. **e**, Epifluorescence images of BALF Mac isolated from WTM (*n* = 15) and OM (*n* = 10) ≥ 221 d p.i., cultured for 12 h, and stained with DAPI, phalloidin, spike and NSP3 Abs. Scale bars, 10 µm. **f**, Spike protein MFI in BALF Mac isolated from HC (*n* = 6), WTM (*n* = 15) and OM (*n* = 10) ≥ 221 d p.i., measured at 2, 4, 8, 12 and 18 h of culture, with HC reference. **g**, Percentage of connected BALF Mac as in **f** measured 12 h postculture initiation. In all graphs, median and interquartile range are shown. Statistical analyses: Mann–Whitney test (**a** and **g**) and Kruskal–Wallis test with Dunn's post hoc test (**c** and **f**). Linear regression is represented in Spearman correlation analyses. Symbols represent individual macaques.

of other immune cell types among total BALF CD45⁺ cells, including CD3⁺CD8⁺ T cells, CD3⁺CD4⁺ T cells, CD20⁺ B cells, CD3⁺CD4⁺CD8⁺ T cells or Lin⁻HLA⁻DR⁺ cells, in both WTM and OM compared to HC (Extended Data Fig. 4f). When we stratified WTM and OM based on SARS-CoV-2 viral load in total BALF cells, those with high viral loads (WTMʰⁱ and OMʰⁱ) displayed an elevated frequency of NKG2R⁺ NK cells (7.03 ± 1.86%) compared to HC (3.59 ± 1.18%; Fig. 4b). In contrast, WTMⁿᵉᵍ and OMⁿᵉᵍ macaques exhibited a higher frequency of NKG2R⁺CD8⁺ T cells (12.3 ± 5.11%) compared to HC (3.06 ± 1.42%; Fig. 4b).

To delve deeper into the functional profile of NKG2R⁺ cells, we assessed IFN-γ, GzmB and CD107a using a specialized flow cytometry panel. The frequency of IFN-γ⁺ NKG2R⁺ NK cells in WTMʰⁱ and OMʰⁱ BALF was lower than in HC, while the frequency of IFN-γ⁺ NKG2R⁺ CD8⁺ T cells was higher in WTMⁿᵉᵍ and OMⁿᵉᵍ compared to HC (Fig. 4c). In contrast, the frequencies of GzmB⁺ and CD107a⁺ NKG2R⁺ NK cells and NKG2R T cells were increased in WTMʰⁱ and OMʰⁱ compared to HC or WTMⁿᵉᵍ/OMⁿᵉᵍ (Fig. 4d,e). In line with this, the frequency of GzmB⁺ and CD107a⁺ NKG2R⁺ NK and/or NKG2R⁺CD8⁺ T cells positively correlated with the frequency of Spike⁺ Mac, while the frequency of IFN-γ⁺ NKG2R⁺ NK cells and/or NKG2R⁺CD8⁺ T cells exhibited a negative correlation with the frequency of Spike⁺ Mac (Fig. 4f). Our analysis of lymphocyte phenotypes in BALF suggested a

link between lower expression of IFN-γ and persistence of SARS-CoV-2 in BALF Macs.

## IFN-γ restrains SARS-CoV-2 replication in Macs

To gauge the impact of IFN-γ on viral replication in Macs, we cultured BALF Mac isolated from all WTM, OM and HC with 100 U ml⁻¹ of rIFN-γ and assessed SARS-CoV-2 dsRNA levels in the Macs at 8 h of culture, before any viral cytopathic effects could emerge. IFN-γ treatment led to a significant reduction in viral dsRNA levels in treated WTM and OM compared to untreated counterpart Macs (Fig. 5a and Extended Data Fig. 5a,b), indicating effective inhibition of SARS-CoV-2 replication in BALF Mac by rIFN-γ over this short period. Notably, IFN-γ treatment also improved WTM and OM Mac cell survival at 12 h after culture initiation (Fig. 5b). There was a negative correlation between the expression of IFN-γ in BALF NK cells in vivo and the frequency of Spike⁺ Mac in BALF (Fig. 5c), further highlighting the impact of IFN-γ on viral persistence in BAL Mac.

IFN-γ treatment enhanced the cell-surface expression of MHC-E, a highly conserved nonclassical MHC class Ib molecule, on BALF Mac, whereas untreated Mac primarily stored MHC-E molecules within intracellular compartments, as expected (Fig. 5d and Extended Data Fig. 5c). Furthermore, IFN-γ treatment resulted in increased total

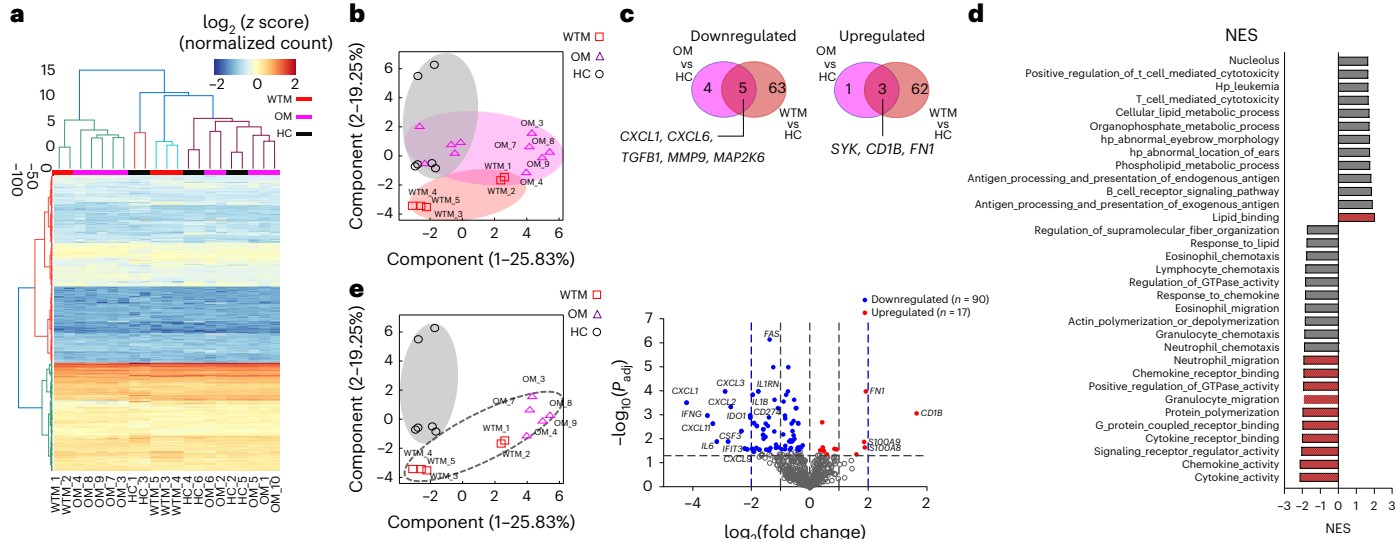

**Fig. 3 | SARS-CoV-2 replication in Mac in vitro alters Mac transcriptomic profile. a**, Heatmap for mRNA expression in BALF CD64⁺ Mac isolated from 5 HC, 5 randomly chosen WTM and 10 OM > 221 d p.i., cultured for 8 h and profiled with the nanoString nCounter System. Cluster distance values were determined with the Euclidean method. **b**, PCA of gene expression in BALF macrophages as in **a**. **c**, Venn diagram showing the genes commonly downregulated and upregulated in BALF MAC (as in Fig. 2a) of WTM and OM compared to HC counterpart. **d**, GSEA hallmark analysis showing enriched gene sets based on the differentially regulated genes from Fig. 2a. Red bars indicate significant enrichment at FDR < 25%, gray bars represent gene sets with FDR > 25% and a nominal $P$ value < 5%. A positive NES value indicates enrichment in WTM and OM, and a negative NES indicates enrichment in HC. **e**, PCA and volcano plots showing upregulated and downregulated genes for the indicated comparison. In each volcano plot, the horizontal dotted line represents a $P_{adj}$ = 0.05 and the vertical dotted line represents a log(fold change) >2 or <−2. NES, normalized enrichment score.

MHC-E expression in WTM and OM Mac compared to untreated BALF Mac counterparts (Fig. 5e and Extended Data Fig. 5d). The mean fluorescence intensity (MFI) of MHC-E on BALF Mac positively correlated with the frequency of IFN-γ⁺NKG2R⁺CD8⁺ T cells in BALF (Fig. 5f), suggesting that IFN-γ likely induced MHC-E expression on Macs in vivo. These findings collectively indicated that IFN-γ inhibited SARS-CoV-2 production and induced MHC-E expression on BALF Mac.

## SARS-CoV-2 infection impacts the NK cell transcriptome

MHC-E interacts with the NK cell receptors NKG2A and NKG2C[20]. This interaction typically inhibits NK cells through NKG2A and activates them through NKG2C[21,22]. Because specific antibodies for NKG2C in macaques are unavailable and existing NKG2A antibodies cross-react with NKG2C on simian cells[23], distinguishing between the NKG2A and NKG2C in macaques generally relies on their mRNA sequences. To investigate the impact of heightened MHC-E expression on NK cell activity, we isolated NKG2R⁺ NK cells from BALF in WTM and HC at approximately 221 d p.i. and analyzed the in vivo transcriptome of 753 NHP immune response-related genes, including KLRC1 (NKG2A) and KLRC2 (NKG2C) mRNA, using the NanoString nCounter codeSet for macaques, and compared them to HC (Fig. 6a). Transcriptomic analysis indicated that BALF NKG2R⁺ NK cells consistently showed low KLRC2 (NKG2C) transcript level compared to KLRC1 (NKG2A) transcript level in both WTM and HC (Fig. 6a).

Based on gene expression, PCA analysis showed that BALF NKG2R⁺ NK cells from 14 of 15 WTM distinctly separated from BALF NKG2R⁺ NK cells in HC (Fig. 6b). Using a significance threshold of adjusted $P$ value ($P_{adj}$ = 0.05) and mRNA log₂(fold change) > 2, only *FOXJ1*, a known inhibitor of nuclear factor kappa B (NF-κB) activation and IFN-γ secretion in T cells[24], was significantly upregulated in BALF NKG2R⁺ NK cells from WTM compared to HC (Fig. 6b). With less stringent criteria, including a nonadjusted $P$ value ($P$ = 0.05) and log₂(fold change) > 1 the comparison between WTM and HC, we identified several genes (*CD36, GzmB, GNLY, PIGR, CD38, SERPING1, KCNJ3, KLRC3, CLU, C3, MUC1* and *FOXJ1*)

that displayed differential modulation in NKG2R⁺ NK cells from WTM compared to HC (Extended Data Fig. 6a).

Using the NanoString nCounter codeSet for analyzing genes expressed within NK cells isolated from the blood and BALF of WTM, we observed that BALF NKG2R⁺ NK cells formed a distinct cluster, clearly separated from blood-derived NK cells, as confirmed by PCA (Fig. 6c,d). In total, 448 genes were differentially expressed (208 upregulated and 240 downregulated) in BALF compared to blood NK cells (Fig. 6d). Notably, mRNA for NK cell receptors (*KLRD1, KLRC3, CD247, KLRB1* and *KLRK1*), transcription factors regulating NK cell development and maturation (*TCF1, ID2, T-BET, EOMES, PRDM1* and *GATA3*) and cytotoxic activity genes (GzM family members, *PERF* and *IFNG*) were downregulated in BALF NK cells compared to blood NK cells in WTM (Fig. 6e). Conversely, mRNA coding for *CD74*, the macrophage inhibitory factor receptor that inhibits NK cell activity, Fcγ receptors, toll-like receptors (TLRs), integrin, pro-inflammatory molecules and the transcription factor IRF8, linked to adaptive NK cells[25], were upregulated in NKG2R⁺ NK cells from BALF compared to blood (Fig. 6e). These findings suggested that BALF NKG2R⁺ NK cells in WTM may have impaired capacity to control SARS-CoV-2 replication, and these transcriptomic changes persisted even after a median of 461 d p.i. The transcriptomic profiles also showed increased *TLR2* and *TLR4* mRNA in NKG2R⁺ BALF NK cells from WTM compared to HC (Fig. 6e).

To further evaluate how NK cells are impacted in the WTM and OM, we used a 20-color marker flow cytometry analysis to assess markers of NK cell activity in BALF NKG2R⁺ NK cells (Fig. 6f and Extended Data Fig. 6b–e). BALF NKG2R⁺ NK cells from WTM exhibited higher expression of NKG2R, CD16 and GzmB compared to OM (Extended Data Fig. 6b). NKG2R expression on bulk NK cells correlated with the percentage of Spike⁺ Mac in BALF (Extended Data Fig. 6c). When clustering macaques according to viral load in total BALF cells, WTM^hi and OM^hi had higher expressions of NKG2R, NKP30, CD16, GzmB and CD107a, but lower expression of CD226 (DNAX Accessory Molecule-1 (DNAM-1)) compared to WTM^neg and OM^neg (Fig. 6g). The changes observed in NKG2R and

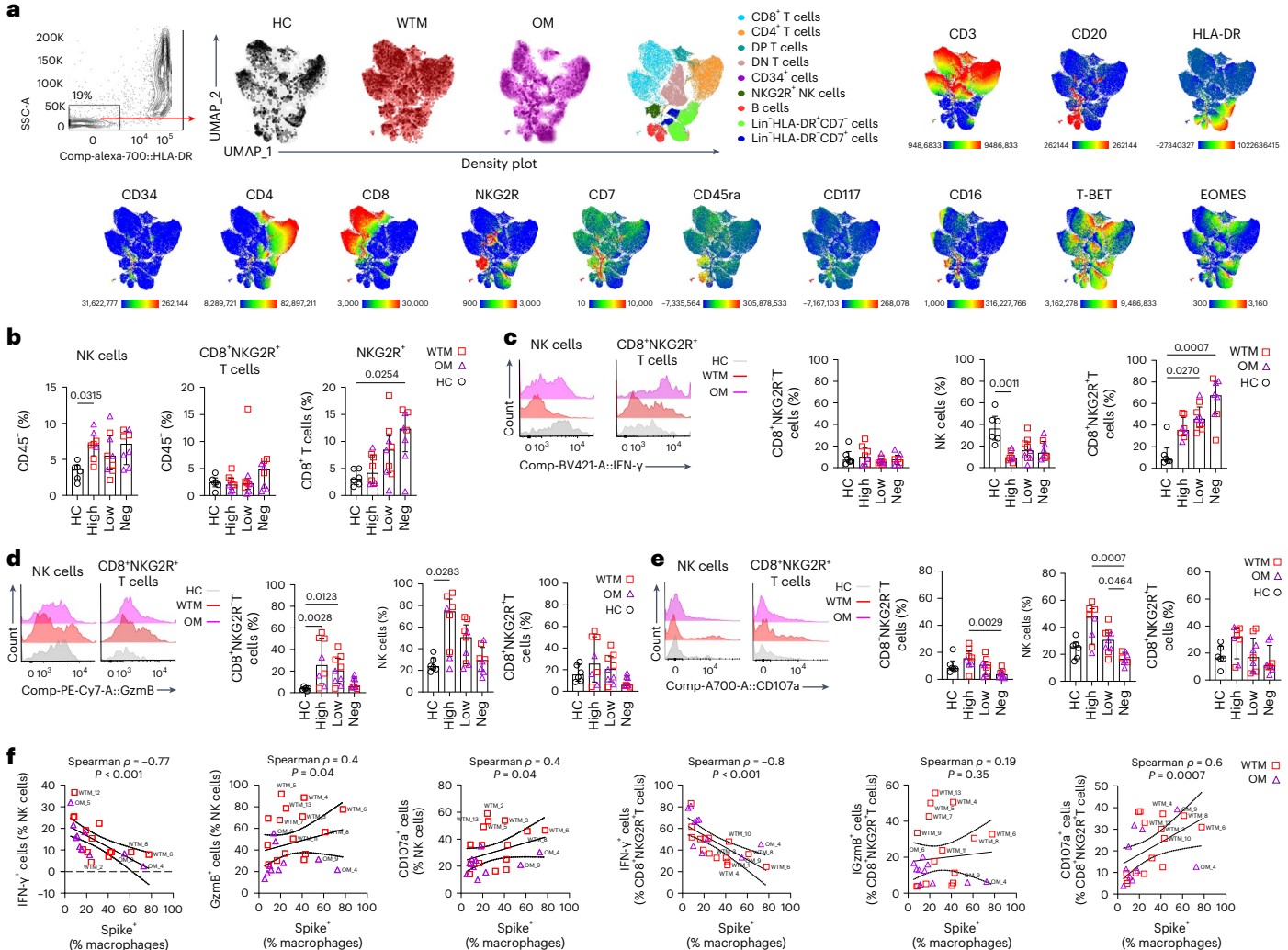

**Fig. 4 | Frequency of IFN-γ⁺ NKG2R⁺ cells negatively correlates with frequency of Spike⁺ Mac. a**, Unsupervised analysis by the UMAP dimension reduction algorithm of Singlet, live, CD45⁺CD14⁻ lymphocytes isolated from BALF of WTM (n = 15), OM (n = 10) at >221 d p.i. and HC (n = 6). Cells were gated, downsampled to 3,000 cells per sample, barcoded and concatenated. CD45 was excluded from the list of UMAP running parameters. Density plot for each group of monkeys is shown. Manually gated lymphocyte populations shown on the UMAP plot with the corresponding color; intensities of each marker used in the analysis shown on the UMAP plots. **b**, Frequency of NKG2R⁺ lymphocyte in BALF cells of WTM, OM and HC at least 221 d p.i. **c–e**, Histograms (left) and frequency (right) of IFN-γ (**c**), granzyme B (**d**) and CD107a (**e**) in NKG2R⁻ T cells, NKG2R⁺ NK cells and CD3⁺NKG2R⁺ T cells from the BALF of WTM, OM and HC as in **a**. **f**, Spearman correlation analysis of the frequencies of IFN-γ⁺, GzmB⁺, CD107a⁺ cells among NKG2R⁺ cells and frequencies of Spike⁺ Mac in BALF cells. In all graphs, the median and the interquartile range are shown. P values were determined using a Kruskal–Wallis test with Dunn's post hoc test. In the correlation analyses, the black solid line represents linear regression, and the dotted lines represent the confidence interval (95%).

CD16 expression, along with differences in GzmB expression, strongly indicated alterations in NK cell maturation.

Next, analysis of BALF NK cell maturation profiles based on CD16 and NKG2R expression indicated that both WTM and OM had lower frequencies of mature NKG2R^lo^CD16⁺ NK cells in BALF compared to HC (Fig. 6h). When clustered based on viral load in total BALF cells, bulk NK cells in WTM^hi^, OM^hi^, WTM^lo^ and OM^lo^ had a decreased frequency of NK cell subsets with low NKG2R expression (NKG2R^lo^CD16⁻ and NKG2R^lo^ CD16⁺) compared to HC (Fig. 6i). The frequency of NKG2R^lo^CD16⁻ and NKG2R^lo^CD16⁺ subsets among total NK cells displayed, negative correlations with the percentage of Spike⁺ Mac in BALF (Extended Data Fig. 6e), suggesting that viral RNA detection in BALF correlated with changes in NK cell phenotypes, population composition and cytolytic function. Overall, the transcriptomic and phenotypic profiles indicated a profound, long-lasting impact of SARS-CoV-2 infection on NK cells, potentially reflecting a blockade of NK cell antiviral activity and an adaptive NK cell profile.

## SARS-CoV-2 spike modulates NK cell function

Next, we analyzed if SARS-CoV-2 particles could directly modulate NK cells. SARS-CoV-2 N and spike proteins engage TLR2 and TLR4 (refs. 26, 27), respectively. TLR2 and TLR4 are known to be expressed on NK cell surfaces[28] and were increased on NK cells in the WTMs. To examine whether BALF NK cells could bind SARS-CoV-2 spike protein, we incubated total BALF cells from WTM, OM and HC with biotinylated recombinant trimeric spike protein (rtSpike) for 1 h (Extended Data Fig. 7a,b). BALF CD20⁺ B cells from OM exhibited strong SARS-CoV-2 spike protein binding (19.65 ± 12%) compared to HC (2.79 ± 0.87%); (Extended Data Fig. 7c). WTM and OM BALF Spike⁺CD20⁺ B cell levels correlated with the titer of RBD IgG in plasma (Extended Data Fig. 7c), suggesting that CD20⁺ B cells could effectively bind trimeric spike protein, possibly as a result of the emergence of SARS-CoV-2-specific memory B cells following the acute phase of the infection[29]. NK cells showed robust binding to spike protein binding (Extended Data Fig. 7b) with a median percentage of 25.8% (±8.78), 23.3% (±14.61) and 16% (±13.87) for

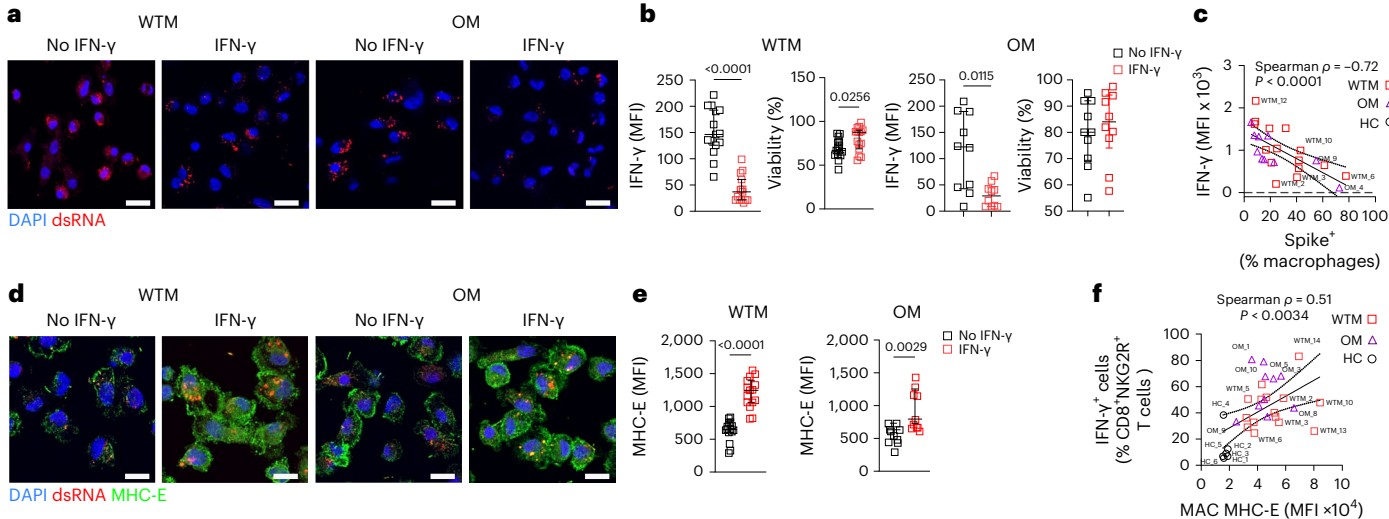

**Fig. 5 | IFN-γ inhibits SARS-CoV-2 replication and increases MHC-E expression. a**, Representative confocal image of BALF Mac from WTM ($n = 15$) and OM ($n = 10$) isolated at least 221 d p.i. and cultured with or without IFN-γ for 8 h, then stained with DAPI and dsRNA Abs. **b**, MFI of dsRNA per cell (left) and viability (right) of BALF Mac as in Fig. 2d. Fluorescence intensity of dsRNA in each cell measured after cell segmentations. Cell viability measured by Trypan blue exclusion. **c**, Spearman correlation analysis of frequency of IFN-γ⁺ NKG2R⁺ NK cells and Spike⁺ Mac in BALF of WTM and OM at the median of 221 d p.i. **d**, Representative confocal image of BALF Mac from WTM and OM isolated

at least 221 d p.i. and cultured with or without IFN-γ for 8 h, then stained with DAPI, dsRNA and MHC-E Abs. **e**, MFI of MHC-E per cell of BALF Mac as in Fig. 2j. Fluorescence intensity of dsRNA in each cell measured after cell segmentations. **f**, Spearman correlation analysis of MHC-E MFI in BALF Mac measured as in Fig. 3k and frequency of IFN-γ⁺CD8⁺NKG2R⁺ T cells measured in BALF of WTM and OM. In all graphs, the median and the interquartile range are shown. $P$ values were determined using a Kruskal–Wallis test. In all correlation analyses, not all animals are labeled, the black solid line represents linear regression and the dotted lines the confidence interval (95%).

---

NKG2R⁺ NK cells from HC, WCM and OCM, respectively (Extended Data Fig. 7d), and thus lower than binding to B cells, but notably higher than the binding to CD4⁺ or CD8⁺ T cells (Extended Data Fig. 7d).

To assess the impact of spike binding to NK cells on NK cell function, BALF NKG2R⁺ NK cells were exposed to rtSpike, monomeric S1 (containing the RBD), trimeric S2 domains, Dimethyl sulfoxide (DMSO) (negative control) or LPS (positive controls) for 24 h (Fig. 7a and Extended Data Fig. 8a). The frequency of GzmB⁺NKG2Rʰⁱ and IFN-γ⁺NKG2Rˡᵒ NK cells increased when WTM or OM BALF NK cells were exposed to rtSpike compared to DMSO (Fig. 7b,c). IFN-γ production was primarily observed in NKG2Rˡᵒ NK cells ($16.4 \pm 10.14\%$), while GzmB production was mainly attributed to NKG2Rʰⁱ NK cells ($17.2 \pm 10.7\%$) (Extended Data Fig. 8a). Following spike protein stimulation, NKG2Rʰⁱ NK cells from WTMⁿᵉᵍ and OMⁿᵉᵍ exhibited higher expression of GzmB than NKG2Rʰⁱ NK cells from WTMʰⁱ and OMʰⁱ, while NKG2Rˡᵒ NK cells from WTMⁿᵉᵍ and OMⁿᵉᵍ had higher expression of IFN-γ compared to NKG2Rˡᵒ NK cells from WTMʰⁱ and OMʰⁱ (Fig. 7d). These observations indicated that NK cells could bind SARS-CoV-2 spike protein and that this affected their function, particularly the production of GzmB and IFN-γ in the NKG2Rʰⁱ and NKG2Rˡᵒ NK cell subsets, respectively.

### MHC-E restricts NK cells in convalescent macaques

MHC-E, the ligand of NKG2R, primarily binds peptides from MHC class Ia signal peptides (SP)[30–32] and can also bind pathogen-derived peptides

like SARS-CoV-2 peptides[16,33]. To explore how NKG2R-MHC-E modulated NK cell activity, we searched for SARS-CoV-2-derived peptides resembling canonical MHC-E binding peptides[34]. In silico analyses considered SP-derived peptide cleavage by protease signal peptide peptidase (SPP; Extended Data Fig. 9a). The nonamer peptide, $V_{3-11}$, was identified within spike protein (position 3–11) and reported to be presented by HLA-A02:01 in vivo[35] (Extended Data Fig. 9b,c). Some viral peptides can bind both classical MHC-I and nonclassical MHC-E molecules, which is potentially relevant in transporter associated with antigen processing (TAP)-deficient viral infections where MHC-E-bound peptides resemble those binding to HLA-A02:01 (refs. 35,36). Despite low binding score predictions (Extended Data Fig. 9d,e), a MHC-E stabilization assay using $V_{3-11}$ peptides on K562 cells transduced with MHC-E01:01 (K562-E01) indicated that $V_{3-11}$ stabilized MHC-E at the cell surface, akin to VL9 and HSP60 peptides (Fig. 8a,b).

We then analyzed the NK cell activity against target cells presenting $V_{3-11}$ peptide through MHC-E. Total Lin⁻CD56⁺CD16⁺ NK cells were isolated from frozen human peripheral blood mononuclear cells (PBMCs) collected before 2019 to ensure donors were SARS-CoV-2 naïve (Fig. 8c and Extended Data Fig. 9f). In cocultures with K562-E*01 pulsed with 30 or 300 μM $V_{3-11}$ peptide, NK cell degranulation, assessed by CD107a expression 8 h later, was compared to that after pulsing with VL9 (known to inhibit NKG2A⁺ NK cells) and HSP60 (no NKG2A⁺ inhibition) control peptides. As expected, VL9 inhibited

---

**Fig. 6 | Expression of *FoxJ1* is upregulated in BALF NKG2R⁺ NK cells. a**, Heatmap of gene expression profiles in NKG2R⁺ NK cells from BALF of 15 WTM and 5 HC at ≥461 d p.i., based on mRNA counts normalized $z$ scores via the nanoString nCounter System. **b**, PCA (top) and volcano plot (bottom) showing differentially expressed genes in BALF NKG2R⁺ NK cells, as in Fig. 4a. Dotted lines indicate significance thresholds ($P_{adj} = 0.05$; log(fold change) >1 or < −1). **c**, Heatmap of gene expression in BALF compared to blood NKGR⁺ NK cells from 12 paired WTM samples based on $z$ scores as in Fig. 4a. **d**, PCA (top) and volcano plot (bottom) showing genes upregulated or downregulated between blood and BALF NKGR⁺ NK cells as in Fig. 4c. Dotted lines indicate significance thresholds. **e**, Heatmaps

of significantly altered mRNAs for various markers and receptors in specific cell populations and tissues. **f**, UMAP analysis of CD45⁺CD14⁻CD3⁻CD20⁻NKG2R⁺ lymphocytes from BALF of WTM, OM and HC. Manually gated lymphocyte populations are color-coded. **g**, MFI of surface markers and cytokines in BALF NKG2R⁺ NK cells from WTM, OM and HC at ≥461 d p.i. **h,i** Contour plots (**h**) and frequency (**i**) of distinct NKG2R/CD16 NK cell subpopulations in HC and WTM (as OM) at ≥461 d p.i. Cluster distance values are determined with the Euclidean method. For all graphs, symbols represent individual macaques; bars indicate medians. Interquartile ranges are shown. $P$ values determined by Kruskal–Wallis test with Dunn's post hoc test.

NKG2A+ NK cells more than NKG2C+NKG2A−NK cells, whereas HSP60 inhibited NKG2C+ and NKG2A− NK cells similarly (Fig. 8d). NKG2C+ NK cell degranulation was not inhibited by V$_{3-11}$, while NKG2C−NKG2A+ NK cell degranulation was (Fig. 8d). Inhibition of NKG2C−NKG2A+ NK

cell degranulation by V$_{3-11}$ was similar to that of VL9 and higher than that of HSP60 (Fig. 8e).

Degranulation activity of BALF NKG2R+ NK cells from HC macaques was similarly inhibited by VL9 and V$_{3-11}$, not by HSP60 (Fig. 8f). However,

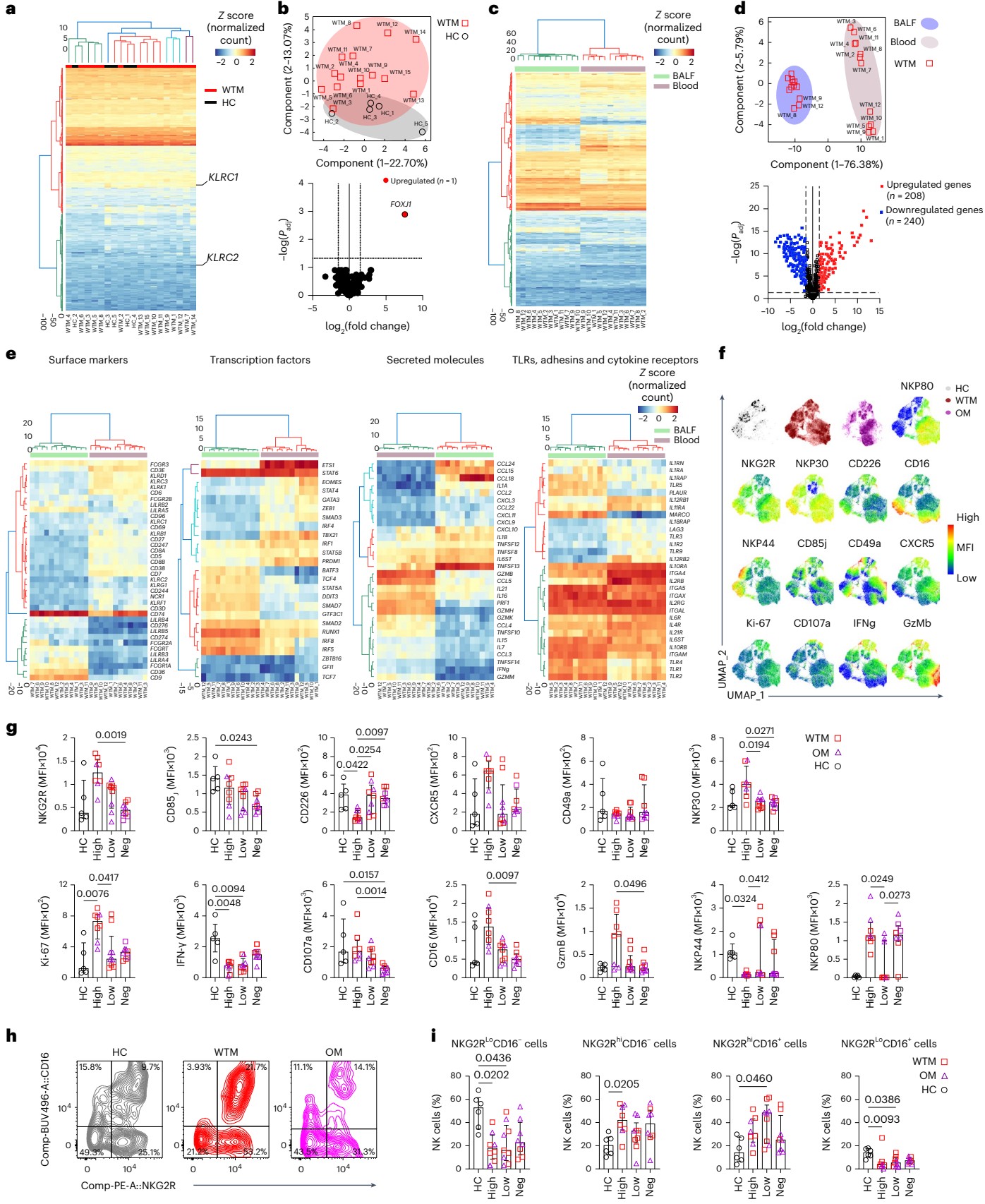

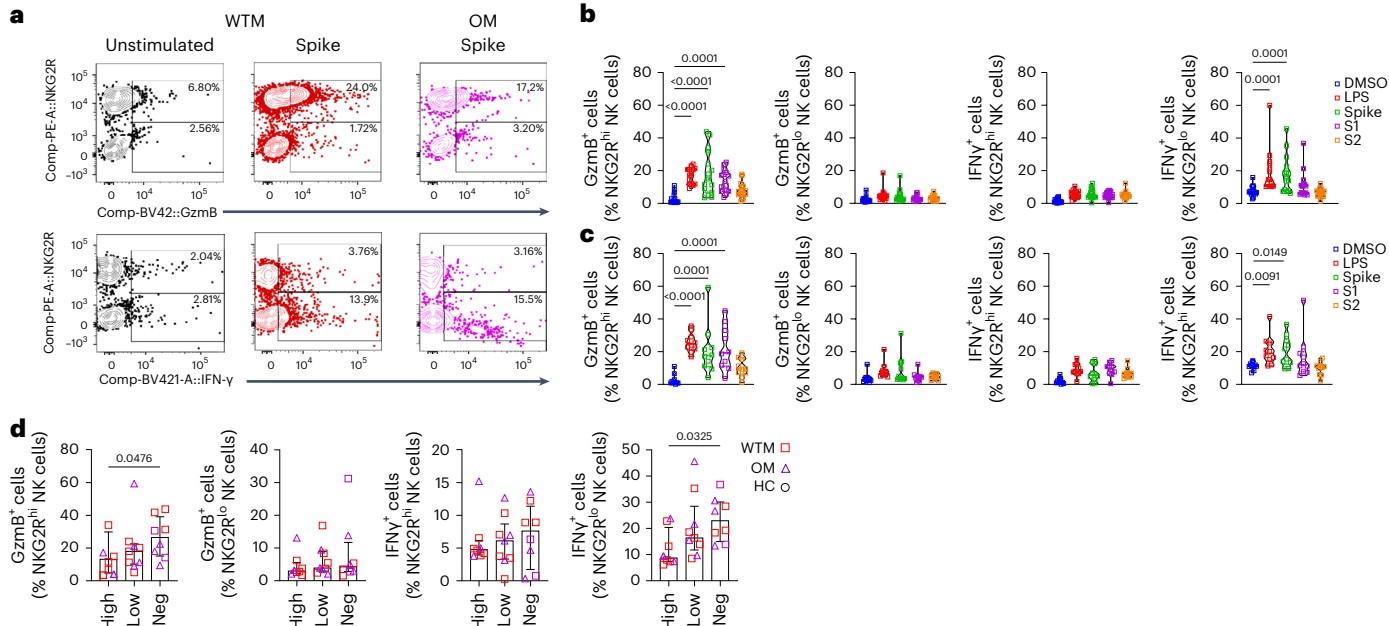

**Fig. 7 | NK cell IFN-γ production following spike stimulation. a**, Representative GzmB and IFN-γ expression plots in NKG2R^lo and NKG2R^hi NK cells from WTM (*n* = 15) and OM (*n* = 10), isolated ≥221 d p.i., cultured with or without spike protein. **b,c**, GzmB and IFN-γ expression in NKG2R^hi and NKG2R^lo NK cells from WTM (**b**) and OM (**c**), isolated ≥221 d p.i., cultured with DMSO, LPS, trimeric spike,

spike domain S1 or spike domain S2. **d**, GzmB and IFN-γ expression in NKG2R^hi and NKG2R^lo NK cells from WTM^neg, WTM^lo, WTM^hi, OM^neg, OM^lo and OM^hi, isolated ≥221 d p.i., and cultured with spike protein for 24 h. Each symbol represents an individual; bars represent medians. Interquartile ranges are shown. In all graphs, *P* values were determined by Kruskal–Wallis test with Dunn's post hoc test.

$V_{3-11}$ inhibited the degranulation of BALF NKG2R^+ NK cells from WTM less than VL9 (Fig. 8f). $V_{311}$ strongly inhibited the degranulation of BALF NKG2R^+ NK cells from WTM^hi, but not that of BALF NKG2R^+ NK cells from WTM^neg (Fig. 8f). Inhibition by $V_{3-11}$ positively correlated with the frequency of Spike^+ Mac and with the amount of viral RNA in total BALF cells from WTM (Fig. 8g), suggesting that NKG2R^+ NK cells from WTM^lo and WTM^neg as well as OM^lo and OM^neg evaded MHC-E-mediated inhibition. Thus, the spike protein-derived $V_{3-11}$ peptide strongly inhibited the MHC-E-dependent degranulation activity of NKG2R^+ NK cells from WTM^hi and OM^hi, whereas NKG2R^+ NK cells from WTM^lo, WTM^neg, OM^lo and OM^neg were less inhibited.

## Discussion

In this study, using Wuhan or Omicron variants of SARS-CoV-2, we detected viral RNA in BALF of 17 of 25 infected macaques and viral antigens in the BALF Mac of 20 of 25 macaques at 6 months p.i., with the virus in the BALF Mac capable of replication up to at least 221 d p.i., even when it was undetectable in nasopharyngeal and tracheal swabs. BALF Mac from WTM^hi and OM^hi exhibited reduced IFN-γ production when stimulated with LPS, and BALF NKG2R^+ NK cells from WTM and OM displayed decreased IFN-γ expression compared to HC. The suppressed IFN-γ expression in BALF NKG2R^lo NK cells could be rescued through stimulation with SARS-CoV-2 spike protein. The spike protein-derived peptide $V_{3-11}$ bound to MHC-E and inhibited NK cell degranulation activity, and the extent of inhibition induced by $V_{3-11}$ correlated significantly with viral load and the presence of Spike^+ Mac in the BALF. NK cells from WTM^lo and OM^lo had a higher degree of resistance to the inhibitory effects of $V_{3-11}$.

WTM had higher viral load in acute infection compared to OM, suggesting infection with Wuhan and Omicron strains mimicked SARS-CoV-2 acute infection in humans[37]. Several other parameters paralleled human findings, such as persistent systemic inflammation months after initial infection[38]. Notably, the frequency of persisting virus in BALF Mac was more elevated in WTM than in OM. Our data demonstrate an active persistent SARS-CoV-2 viral reservoir, consistent

with findings of persisting viral reservoirs in humans[5]. It is worth noting that the frequency of macaques with persisting virus may be higher than in humans, eventually due to our high infection doses.

In vitro studies have shown that SARS-CoV-2 can enter human Mac[9,39,40]. Angiotensin Converting Enzyme 2 (ACE-2) expression has been described in Mac from patients with COVID-19 and viral proteins have been observed in Macs from various tissues postmortem[4]. This suggests that Macs could be directly infected by SARS-CoV-2 in vivo. Detection of viral antigen and RNA in Mac might also result from phagocytosis of infected alveolar epithelial cells, possibly followed by lysosome escape[41]. Antibody-dependent enhancement of cellular infection may also be a mechanism of Mac infection, as seen in SARS-CoV and Middle East respiratory syndrome coronavirus (MERS-CoV) infection[42]. It's likely that other cellular reservoirs in the lung exist, but our data suggest that SARS-CoV-2 exploits Mac as a key repository for dissemination and long-term persistence.

While the entry of SARS-CoV-2 into human monocytes and Mac is documented, the release of infectious viral particles from Mac has remained unclear. Our study shows that SARS-CoV-2 is produced in primary cultures of BALF Mac. Increasing virus levels during culture corresponded to ongoing viral replication and were accompanied by virus-induced morphological changes in the cells and gene expression changes. The morphological changes include membrane ruffling and the presence of filiform extensions that connected multiple Mac in culture, with viral proteins detected within these extensions. This suggests virus dissemination through cell-to-cell mechanisms, aligning with reports on tunneling nanotubes in SARS-CoV-2 spreading[43]. Such mechanisms could allow the escape of immune system sensing and provide a further means for SARS-CoV-2 to persist in tissues. In some macaques, SARS-CoV-2 production was detected after BALF Mac culture even when viral RNA in BALF was undetectable ex vivo, suggesting the presence of occult infection that can be re-activated.

Transcriptomic profiles of cultured BALF Macs revealed increased expression of *FN1* (fibronectin), potentially contributing to the observed morphological changes. Expression of S100A8/S100A9

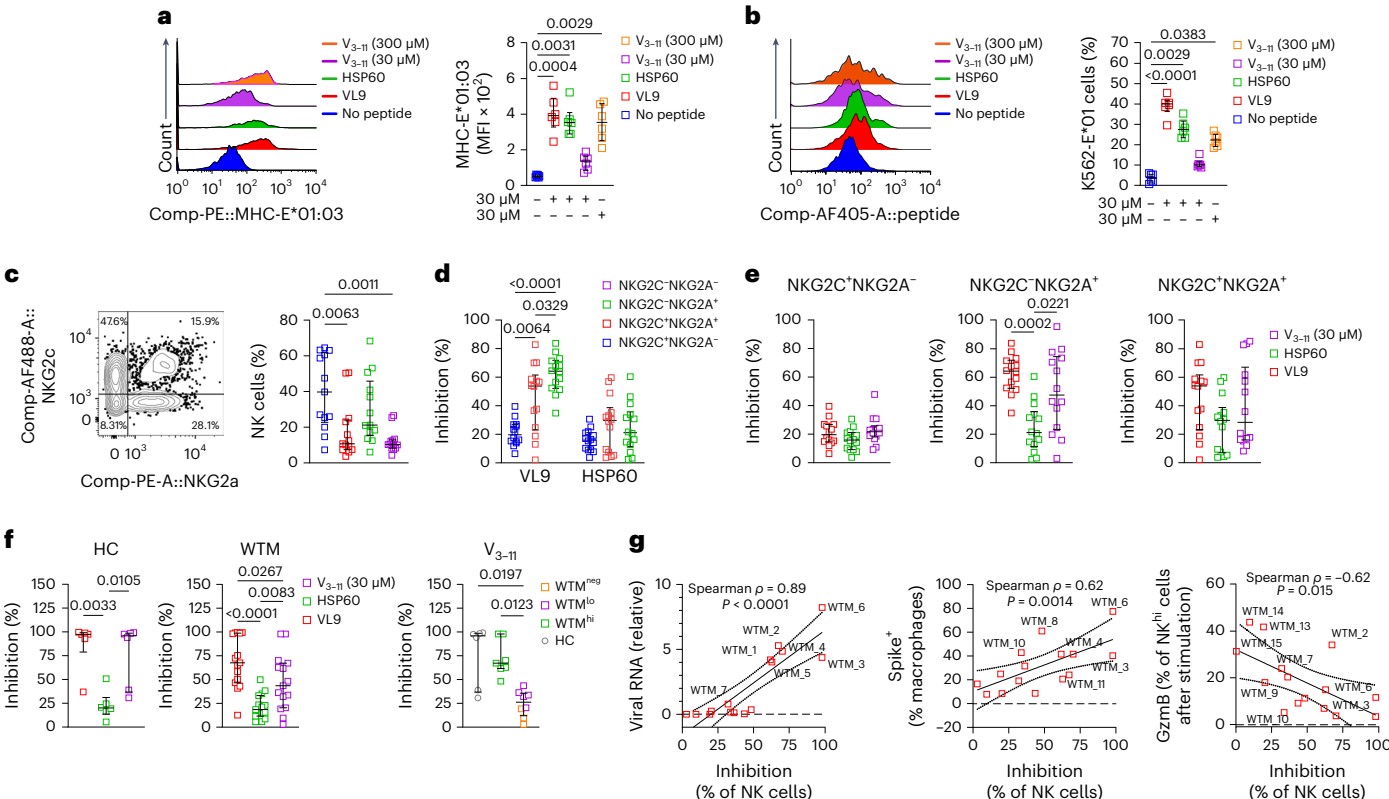

**Fig. 8 | Spike leader sequence peptide inhibits NK cell lysis. a**, Histogram showing the MFI of MHC-E in K562 cells transduced with MHC-E (MHC-E*01:03) after peptide loading (VL9, HSP60, $V_{3-11}$) and culture for 12 h (left), and graph illustrating the expression of MHC-E in MHC-E*01:03 cells after peptide loading (with VL9, HSP60, and $V_{3-11}$ peptides) compared to the control condition with no peptides, following a 12-h culture (right). **b**, Histogram displaying the MFI of biotinylated peptides (VL9, HSP60, $V_{3-11}$) loaded into MHC-E*01 cells and revealed through conjugation with streptavidin, following a 12-h culture period (left), and expression of biotinylated peptides (VL9, HSP60, $V_{3-11}$) in MHC-E*01:03 cells after peptide loading (with VL9, HSP60 and $V_{3-11}$ peptides), in comparison to the control condition with no peptides, following a 12-h culture period (right). **c**, Dot plot illustrating the distribution of blood NK cells from 13 human PBMCs collected before 2019 based on NKG2A and NKG2C expression (left) and frequency of NKG2C$^+$NKG2A$^-$, NKG2C$^+$NKG2A$^+$ and NKG2C$^-$NKG2A$^+$

cell subsets in human PBMCs collected before 2019 (right). **d**, Inhibition of degranulation in human NKG2C$^+$NKG2A$^-$, NKG2C$^+$NKG2A$^+$, NKG2C$^-$NKG2A$^+$ and NKG2C$^-$NKG2A$^-$ NK cell subsets exposed to K562-E-01 cells loaded with VL9 and HSP60 peptides during an 8-h coculture, as in **c**. **e**, Inhibition of degranulation in human NKG2C$^+$NKG2A$^-$, NKG2C$^-$NKG2A$^+$ and NKG2C$^+$NKG2A$^+$ NK subsets during 8-h coculture with VL9, HSP60 and the spike-derived peptide $V_{3-11}$. **f**, Inhibition of degranulation in BALF NKG2R$^+$ NK cells isolated ≥461 d p.i. from 15 WTM and incubated with $V_{3-11}$ peptide-loaded K562-E-01 cells for 8 h. **g**, Spearman correlation analysis between $V_{3-11}$ peptide-induced inhibition of NKG2R$^+$ NK cell degranulation and viral load measured in Fig. 2a, Spike$^+$ Mac frequency measured in Fig. 2c and GzmB expression in NKG2R$^{hi}$ NK cells in total BALF cells of WTM after 24 h of spike stimulation, as measured in Fig. 7b. Each symbol represents an individual; bars represent medians. Interquartile ranges are shown. In all graphs, P values were determined by Kruskal–Wallis test with Dunn's post hoc test.

(calprotectin) was also elevated, similar to reports in infected humans[38]. S100A8/S100A9 has been linked to chronic TLR4/RAGE signaling and chronic expression of IL-1b, IL-6 and tumor necrosis factor in long COVID/Postacute sequelae of SARS-CoV-2 infection (PASC)[44], potentially explaining the elevated IL-6 expression in WTM and OM plasma. CD1b, a nonpolymorphic MHC class I-like glycoprotein that activates NK cells through CD36, was increased in BALF WTM and OM Mac in vitro compared to HC. Antigen processing pathways for CD1 and MHC-E-bound ligands mainly occur in endosomal compartments[32,45], which SARS-CoV-2 can manipulate[46]. The impact of endosome manipulation on CD1 or MHC-E peptide processing remains poorly explored[47]. Our data, showing increased MHC-E and CD1b expression in BALF Mac, suggest upregulation of noncanonical antigen presentation pathways.

We found that IFN-γ inhibited SARS-CoV-2 replication in BALF Mac. IFN-γ expression was reduced in BALF Mac and NK cells in WTM and OM compared to HC. In BALF Mac from convalescent macaques, increased IL-10 expression may inhibit IFN-γ. Transcriptomic analysis indicated increased expression of *FOXJ1* mRNA in WTM and OM BALF NK cells compared to HC BALF NK cells. FOXJ1 is known to inhibit NF-κB activation and IFN-γ secretion in CD4$^+$ T cells[24], but its function in NK cells is unknown. The impairment in IFN-γ production in BALF NK cells

and Mac might be partly compensated by IFN-γ production in unconventional NKG2R$^+$CD8$^+$ T cells. Additionally, IFN-γ production by more mature NKG2A$^{lo}$ NK cells might be stimulated by spike protein. IFN-γ also induced the upregulation of MHC-E on BALF Mac. Because MHC-E inhibits NKG2A$^+$ NK cells, its upregulation might protect the Mac from NK cell-mediated lysis. Thus, IFN-γ inhibited viral replication, but concomitantly rendered the SARS-CoV-2-infected cells more resistant to NK cell killing through MHC-E upregulation. Thus, IFN-γ might promote viral persistence by inhibiting the elimination of infected cells, despite the fact that it restrains viral replication.

NK cell degranulation activity was inhibited by the $V_{3-11}$ peptide presented through MHC-E, suggesting a role for this spike peptide in SARS-CoV-2 evasion of the immune response. Peptides from SARS-CoV-2 spike and Nsp13 proteins can bind to MHC-E[16,33]. Target cells presenting $V_{3-11}$ through MHC-E inhibited the degranulation of NK cells from SARS-CoV-2-naïve macaques and humans, but not that of NK cells from WTM$^{neg}$ and OM$^{neg}$. This observation indicated the induction of adaptive NK cells in WTM$^{neg}$ and OM$^{neg}$ that escape MHC-E-mediated inhibition and further highlighted the intricate interplay between Mac and NK cells. It is unclear why NK cells with such adaptive function arouse in some, but not all macaques. The enhanced

MHC-E-restricted NK cell adaptive activity may be related to early infection events, cytokine environments, prior infections, vaccinations and host genetics. In conclusion, our study showed that SARS-CoV-2 persisted in lung Mac and could disseminate through cell-to-cell channels. IFN-γ, produced by NKG2R⁺CD8⁺ T cells and NKG2Rᴸᵒ NK cells, regulated SARS-CoV-2 persistence and induced the expression of MHC-E on Mac. The NK cells from some macaques could evade the MHC-E-mediated inhibition, prompting further investigation into the inducibility of such NK cells by immunotherapies and vaccines.

## Online content

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

## Methods

### Human samples

Blood samples from healthy donors were obtained from the French blood bank (Etablissement Français du Sang) as part of an agreement with the Institut Pasteur (C CPSL UNT, 15/EFS/023). The study was approved by the Ethics Review Committee (Comité de protection des personnes) of Île-de-France VII. Participants' sex, number and age were not available due to the anonymous and nonremunerated nature of the blood donations. Fifty-milliliter blood samples were obtained from 12 donors. All blood samples were collected before 2019, and PBMCs were isolated using Ficoll gradient and immediately frozen.

### Macaques

Cynomolgus macaques (*Macaca fascicularis*) aged 37–60 months were sourced from Mauritian AAALAC-certified breeding centers. The study was conducted at Infectious Diseases Models for Innovative Therapies (IDMIT) infrastructure facilities with appropriate containment levels in accordance with regulations. At the time of infection, all animals were seronegative for specific viruses. Following infection, the macaques were housed individually in biosafety facilities. Both male and female monkeys were included in the study, and sample collection was performed randomly. The study protocols received approval from the institutional ethical committee, 'Comité d'Ethique en Expérimentation Animale du Commissariat à l'Energie Atomique et aux Energies Alternatives' (CEtEA 44), under statement numbers A20_011, 20_066 and 21_069. Authorization for the study was granted by the 'Research, Innovation, and Education Ministry' under registration numbers APAFIS 24434-2020030216532863v1, APAFIS 29191-2021011811505374v1 and APAFIS 33414-2021101115102064v1.

### Blinding and randomization

During sample collection, the investigators were not blinded, but the animal handlers were unaware of group allocations. Data collection procedures were carefully randomized or appropriately blocked to minimize potential bias. We used randomization techniques in the organization of experimental conditions and stimulus presentation, further enhancing the reliability and validity of our findings. All animals involved in our study were included in the research, and none were excluded at any point during the study's duration.

### SARS-CoV-2 strains

For in vivo investigations, following three distinct SARS-CoV-2 isolates were used: the human wild-type (Wuhan) strain (hCoV-19/France/IDF0372/2020 strain), the Omicron BA.1 VOC (hCoV-19/USA/GAEHC-2811C/2021, EPI_ISL_7171744) and the Omicron BA.2 VOC (hCoV-19/Japan/UT-NCD1288-2 N/2022, EPI_ISL_9595604). Virus stocks used in vivo were generated through two passages on mycoplasma-free Vero E6 cells. Virus titration was performed on Vero E6 cells.

### Experimental group size estimation

We followed the '3Rs' principle for animal research, aiming to minimize the use of experimental subjects. Animals were selected to closely match in terms of age, weight and genotype. Sample size calculations were performed using power analysis, which accounts for effect size, s.d., type 1 error and 80% power in a two-sample $t$ test with a 5% significance level (two-sided test). We used G power software version 3.1.9.7. for these calculations, with $D$ determined as (difference in means)/(s.d.), estimated from our preliminary results. We considered a maximum $D$ value of 1.8 based on previous studies for sample size calculation.

### Animal infection protocol

The viral dose and route of infection were administered following previously described protocols[48]. Challenged macaques were exposed to a total dose of $10^6$ PFU of SARS-CoV-2 through a combination of intranasal and intratracheal routes (day 0), using atropine (0.04 mg kg$^{-1}$) as premedication and ketamine (5 mg kg$^{-1}$) with medetomidine (0.042 mg kg$^{-1}$) as anesthesia.

### Tissue collection and processing

PBMCs were isolated using Ficoll density-gradient centrifugation. BALF samples underwent centrifugation, and mononuclear cells were separated using standard density-gradient centrifugation.

### SARS-CoV-2 viral RNA quantification in nasopharyngeal and tracheal samples

Upper respiratory (nasopharyngeal and tracheal) specimens were collected using swabs (universal transport medium, Copan; or viral transport medium, CDC, DSR-052-01). Tracheal swabs were obtained by inserting the swab just above the tip of the epiglottis into the upper trachea, approximately 1.5 cm beyond the epiglottis tip. All collected specimens were stored between 2 °C and 8 °C until analysis. Viral copy numbers were determined via quantitative RT–PCR using a plasmid standard concentration range containing an RdRp gene fragment that included the RdRp-IP4 RT–PCR target sequence. The estimated limit of detection was 460 copies, as previously described[48]. Detailed protocols are also accessible on the World Health Organization (WHO) website (https://www.who.int/docs/default-source/coronaviruse/real-time-rt-pcr-assays-for-the-detection-of-sars-cov-2-institut-pasteur-paris.pdf?sfvrsn=3662fcb6_2). Primer sets sequence nCoV_IP2 (5′-ATGAGCTTAGTCCTGTTG-3′, 5′-CTCCCTT TGTTGTGTTGT-3′) and nCoV_IP4 (5′-GGTAACTGGTATGATTTCG-3′, 5′-CTGGTCAAGGTTAATATAGG-3′).

### SARS-CoV-2 viral RNA quantification in BALF cells

Viral copy numbers were determined via quantitative RT–PCR. Detailed protocols are accessible on the WHO website. Relative quantification of the viral genome was conducted using one-step RT–qPCR with the RNA-to-CT 1 Step Kit (Applied Biosystems, 4392938). Thermal cycling was conducted in a 7500 real-time PCR system (Applied Biosystems) equipped with Windows 10 and 7500 Fast software version 2.3. The $2^{-\Delta\Delta Ct}$ was calculated based on the mean Ct values for viral RNA using the nCoV-IP4 primers and 18S RNA Cts obtained for each monkey.

### SARS-CoV-2 ELISA

ELISAs were performed as previously described[49]. Optical densities were measured at 405 nm, and area under the curve values were calculated.

### Luminex (Bio-Plex) and ELISA assay for plasmatic inflammatory mediators

The Luminex and ELISA assay was used to measure a panel of essential plasmatic anti- and pro-inflammatory mediators including IFN-γ, IL-1β, IL-23, IL-6, IL-15, IP-10, IL-18 and CD14s. The measurement process adhered to the guidelines provided by the manufacturer. Measurements were conducted with the Bio-Plex 200 system.

### Isolation of NK cells and Mac

For the isolation of Human CD56$^+$ NK cells, we achieved a purity of over 90% from PBMC through positive selection using antibody-coated magnetic beads provided by Miltenyi Biotec. Specifically, the anti-NKG2A PE-conjugated monoclonal antibody (clone Z199; Beckman Coulter Life Sciences) was used, and all steps of cell staining and selection were meticulously performed following the manufacturer's instructions.

Similarly, Simian NKG2R$^+$ NK cells were isolated to a purity exceeding 90% from both BALF and blood samples using positive selection with antibody-coated magnetic beads from Miltenyi Biotec. In this case, we also used the anti-NKG2A PE-conjugated monoclonal antibody

(clone Z199; Beckman Coulter Life Sciences), and the entire process of cell staining and selection was strictly carried out in accordance with the supplier's instructions.

For the isolation of Mac from BALF samples, we used a positive selection strategy using the anti-CD64 PE-conjugated monoclonal antibody (clone 10.1; BD Biosciences). To identify the isolated Mac, we used anti-PE microbeads from Miltenyi Biotec. Like the previous isolations, the staining and selection procedures for Mac were conducted following the supplier's guidelines.

## Cell culture

K562 (human HLA class I–negative erythroleukemia) cells (ATCC, CCL-243) and HLA-E transduced K562 cells were cultured in RPMI 1640 (Life Technologies) with 10% heat-inactivated fetal calf serum (FCS), 2 mM L-glutamine, 100 U ml$^{-1}$ penicillin and 100 μg ml$^{-1}$ streptomycin.

NK cells were cultured in RPMI 1640 with Glutamax (Life Technologies) supplemented with 10% heat-inactivated FCS, 2 mM L-glutamine, 100 U ml$^{-1}$ penicillin, 100 μg ml$^{-1}$ streptomycin, 100 IU ml$^{-1}$ IL-2 and 10 ng ml$^{-1}$ IL-15.

Mac, postpositive selection, were cultured at a concentration of $5 \times 10^5$ cells per ml in a 24-well tissue culture plate (Nunc) in RPMI 1640 supplemented with FBS and 10 ng ml$^{-1}$ Granulocyte-Macrophage Colony Stimulating Factor (GM-CSF) for up to 24 h. Macrophage stimulation was carried out at a concentration of $1 \times 10^6$ cells per ml, with options for no treatment, 100 ng ml$^{-1}$ LPS (Sigma-Aldrich) or 20 ng ml$^{-1}$ IFN-γ (R&D Systems) for the specified duration

## NK cell stimulation assays

NK cells were grown in RPMI medium 1640 (supplemented with 10% FBS, 2 mmol l$^{-1}$ L-glutamine, 1% penicillin–streptomycin–neomycin, 2-ME), at a concentration of $2.0 \times 10^6$ cells per ml recombinant human IL-2 (50 U ml$^{-1}$). LPS or spike compounds were added to the culture at the following concentrations: 100 ng ml$^{-1}$ LPS (Sigma-Aldrich) and 10 μg ml$^{-1}$ of spike compounds. The plates were then incubated at 37 °C for 8 h. Cells were then collected, fixed with 4% Paraformaldehyde (PFA) and stained for flow analysis.

## Fluorescence quantification and cell segmentation methodology

In this study, we used a robust methodology for quantifying individual fluorescence in Mac during primary culture. This process, adapted from previous work[50,51], ensured accurate and reproducible results. The following workflow, implemented using Fiji software (version 1.53 with Java1.8.0_322), was used for image segmentation: (1) image reprocessing—images underwent preprocessing using Find Edges filters. This step enhanced the subsequent thresholding process. Minimum and maximum sliders were manually adjusted to optimize image saturation, ensuring clarity and accuracy. (2) Thresholding—it was applied to create a binary mask from the preprocessed image. This step separated desired objects from the background. (3) Mask creation and manipulation—a mask, representing the segmented objects of interest, was created based on the thresholded image. The mask isolated regions of significance for analysis. (4) Selection transfer—selections were transferred from the mask to the original image. This step allowed for the extraction of quantitative data from the selected regions. (5) Data analysis—the 'Analyze Particles' tool was used to extract desirable objects based on the selections. Individual MFI statistics for each channel were calculated and reported for the segmented objects. These data provided insights into fluorescence levels within Mac.

## Antibody against SARS-CoV-2 spike protein

The human monoclonal antibody Cv2.31942 was cloned from a patient convalescent from COVID-19, produced as a recombinant IgG1, and purified[49].

## SARS-CoV-2 protein production

Codon-optimized nucleotide fragments encoding stabilized versions of trimeric SARS-CoV-2 and BA.1 spike ectodomains, followed by C-terminal tags (8xHis-tag and AviTag), were synthesized and cloned into pcDNA3.1/Zeo(+) expression vector (Thermo Fisher Scientific). Glycoproteins were produced by transient transfection of exponentially growing Freestyle 293-F suspension cells (Thermo Fisher Scientific) using polyethylenimine precipitation method. Proteins were purified from culture supernatants by high-performance chromatography using the Ni Sepharose Excel Resin according to the manufacturer's instructions (GE HealthCare), dialyzed against PBS using Slide-A-Lyzer dialysis cassettes (Thermo Fisher Scientific), quantified using NanoDrop 2000 instrument (Thermo Fisher Scientific) and controlled for purity by SDS–PAGE using NuPAGE 4–12% Bis–Tris gels (Life Technologies). For polychromatic flow cytometry immunofluorescence staining, SARS-CoV-2 tri-S proteins were biotinylated using Enzymatic Protein Biotinylation Kit (Sigma-Aldrich).

## Search for SARS-CoV-2 peptides with potential for efficient binding to MHC-E

We used IEDB prediction tools (version 2.27) to identify peptides with the potential to bind to MHC-E. Recent studies have revealed that SARS-CoV-2 NSP13 contains peptides capable of binding to MHC-E[16]. While many of these peptides were initially identified in silico and subsequently validated in vitro, it is important to recognize that MHC-E peptide presentation can occur independently of TAP, particularly in cells with deficiencies in components of the MHC class I processing pathway, such as TAP or Interferon Regulatory Factor 1 (IRF1)[32,52,53]. Such deficiencies may arise during SARS-CoV-2 infection, leading to an alternative HLA-E peptide repertoire[32,54].

Intriguingly, most HLA class I molecules contribute their leader sequences for binding to the nonclassical HLA-E, with the cleavage of these signal sequences mediated by SPas and SPP[31,55,56]. These leader peptides, in fact, represent the dominant source of peptides for HLA-E. The essential intramembrane proteolysis of signal peptides is a crucial step in generating MHC class I signal peptides[32,52]. However, the precise loading mechanism for TAP-independent signal peptides into MHC class I molecules remains elusive[57].

In eukaryotic cells, secretory and membrane proteins contain a signal sequence crucial for guiding protein targeting to the endoplasmic reticulum (ER)[58,59]. Following insertion into the protein-conduction channel, signal peptides are typically cleaved from the preprotein by signal peptidase (SPas)[60]. Subsequently, these signal peptides, confined as small domains within the ER membrane, may undergo intramembrane proteolysis via cleavage within their transmembrane region, facilitated by the presenilin-type aspartic protease SPP[32,61,62]. Given that processing by SPas and SPP is believed to occur outside the peptide-loading complex, this intricate machinery has a pivotal role in optimizing ligand length and quality, thus facilitating peptide loading onto nascent MHC class I molecules. Additionally, it 'edits' the repertoire of bound peptides to maximize their affinity. Peptide ligands suitable for MHC class I binding are thought to be generated following intramembrane proteolysis by SPP, which promotes the release of signal peptide fragments from the ER membrane[32].

Therefore, our investigation led us to explore additional peptides with the potential to bind to MHC-E using nonclassical pathways. In this context, we performed a signal peptide structural analysis using SignalP 6.0 (ref. 63). Using signalp software (version 6.0), we scrutinized the N-terminal region of the spike protein from MERS-CoV (A0A140AYZ5), SARS-CoV-1 (P59594) and SARS-CoV-2 (P0DTC2). This analysis strongly indicates that the signal peptide of the SARS spike protein can be cleaved by signal peptidase (SPas) and SPP. Typically, signal peptide sequences consist of the following three domains: a hydrophobic core (h region) in blue, spanning 6–15 amino acids; a polar C-terminal end (c region) in green, characterized by small uncharged

amino acids; a polar N-terminal region (n region) in red, with a positive net charge. It is also worth noting that Sec/SPI signal peptides, classified as 'standard' secretory signal peptides, undergo transport through the Sec translocon and are cleaved by signal peptidase I. Among these peptides, three have been described in the immune epitope database (IEDB) and experimentally validated. Notably, the $V_{3-11}$ peptide sequence bears the closest resemblance to the canonical MHC-E binding motif.

## Peptides and HLA-E stabilization assay

Synthetic peptides biotinylated or not were purchased from Proimmune and dissolved in DMSO at the concentration of 2 mg ml$^{-1}$. The control peptides used were VL9 (VMAPRTVLL) and HSP60 (QMRPVSRVL). K562 and K562-E*0101 cells were incubated with synthetic peptides (3–300 μM) at 37 °C for 15–20 h in serum-free AIM-V medium (GIBCO BRL) at a concentration of $1-3 \times 10^6$ cells per ml. Control cultures were kept at 37 °C for over 16 h without peptides. Cells were then collected, washed in PBS and cell-surface expression of HLA-E was determined by incubation with PE-conjugated anti-HLA-E antibody, washed twice with PBS and fixed in 100 μl of Cyto x (BD Biosciences). Cells were acquired using a Fortessa (BD Biosciences) equipped with DIVA software (version 8.0), and FlowJo software (version 10.4.2; Tree Star) was used for all analyses. Results were expressed directly as MFI.

## Polychromatic flow cytometry immunofluorescence staining

The antibodies used for flow cytometry and immunofluorescence staining are listed in Supplementary Table 1. PBMCs and BALF lymphocytes were stained as described previously[64], with crystal violet[65] used to minimize alveolar macrophage autofluorescence. The anti-NKG2A antibody recognizes NKG2A, NKG2C and NKG2E on simian cells and NKG2A on human cells. Flow cytometry was conducted using an LSR-Fortessa (BD Biosciences), and intracellular staining was achieved with BD Cyto x/Cytoperm. Data analysis was performed with FlowJo 10.4.2 software (FlowJo, LLC), using FlowJo plugins Phenograph (version 3.0) and uniform manifold approximation and projection (UMAP; version 3.1). Phenograph used a $K$ mean of 50, while UMAP used nearest neighbors with a value of 25 and a minimal value of 0.25.

Immunofluorescence staining involved fixing Mac in 4% PFA after various culture periods, followed by staining. Membrane permeabilization was achieved with Triton X-100, and primary antibodies were applied. After a 2-h incubation, unbound primary antibodies were washed with PBS. Secondary antibodies (Thermo Fisher Scientific; 1:200) coupled with fluorescent dyes were used for detection for 1 h at 37 °C. Sections were mounted with Vectashield containing DAPI (4′6-diamidino-2-phenylindole; Vector Laboratories). Immunofluorescence analysis was conducted using a CELENA X High Content Imaging System (Logos Biosystems) equipped with the CELENA X Explorer (version 1.6.0) or a spinning-disk Ti2E Microscope (Nikon−Yokogawa CSU-W1).

## NanoString NHP immunology V2 panel multiplex gene expression profiling

We conducted gene expression profiling using the NanoString nCounter analysis system (NanoString Technologies) with whole cell lysates in RNA Later Tissue (RLT) buffer (20,000 cells per μl) following supplier instructions. RNA transcripts were analyzed with the nCounter NHP Immunology V2 Panel, encompassing 754 genes and 16 internal reference genes for data normalization. Data analysis used ROSALIND (https://rosalind.bio/) with HyperScale architecture by ROSALIND. Quality control involved read distribution percentages, violin plots, identity heatmaps and sample MDS plots. Normalization was performed by dividing counts within a lane by the geometric mean of normalizer probes from the same lane, selecting housekeeping probes via geNorm. Fold changes and $P$ values were computed using the fast method from the nCounter Advanced Analysis 2.0 User Manual, with

$P$ value adjustment via the Benjamini−Hochberg method for false discovery rates (FDRs). Clustering of differentially expressed genes into the final heatmap used the partitioning around medoids method, considering factors like signal direction, gene position, role and type, among others, with the fpc R library.

## MHC-E-dependent NK cell viral suppressor assays

NK cell activity was determined through the expression of cell surface CD107a, as previously described[66]. K562-E*0101 cells were incubated with 30–300 μM of a given peptide at 26 °C for 15–20 h. NK cells were cocultured with $2 \times 10^4$ K562-E*0101 target cells pulsed or not with peptides at an NK cell/target cell ratio of 5:1. Target cells were in parallel cultured in the absence of NK cells. Anti-CD107a antibody was added at the start of the assay, and GolgiStop and GolgiPlug (both BD Biosciences) were added 1 h after start of the stimulation. After 8 h of culture, cell suspensions were stained for viability and cytotoxic activity was analyzed through measurement of CD107a expression by flow cytometry. The percentage of inhibition was calculated as follows: % of inhibition = 100 − specific lysis. The specific lysis was calculated as described[66,67] using the following formula: $(E - S)/(M - S) \times 100$ (where E = NK cells degranulation against target cells loaded with peptides, S = spontaneous NK cells degranulation and M = NK cells degranulation against target cells without peptides).

## Statistics

Data analysis and statistical tests were conducted using GraphPad Prism 7 (GraphPad Software, version 10.0.2). Various statistical analyses were applied in the study, including the Mann-Whitney Test, utilized in Figs. 1b, 4a, and 7; the Kruskal-Wallis Test with Dunn's post hoc test, employed in Figs. 1c,g,h, 2a,c, 3, 4a,g, 5–7, and 8d,e; the Non-parametric Wilcoxon Signed-Rank Test, applied in Fig. 1k; Linear Regression, used in Figs. 1b, 3d, 5c, 6h,i, and 8f; and Spearman Correlation Analysis, conducted in Figs. 1k, 4f, 5c,f, 6h,i, 7f, and 8l. A significance threshold of $P < 0.05$ was considered. FDRs and $P_{adj}$ corrections were applied for multiple comparisons.

## Reporting summary

Further information on research design is available in the Nature Portfolio Reporting Summary linked to this article.

# Data availability

The authors declare that all other data supporting the findings of this study are available within the article and its raw data files or are available from the authors upon request. Source data are provided with this paper.

# Code availability

The source code and related materials are available for access and download at Gene Expression Omnibus under references GSE243731, GSE243732, GSE243733 and GSE243734 for further examination and utilization.

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

## Acknowledgements

We are grateful for the excellent help from the veterinarians and staff at the IDMIT Center (B. Delache, E. Burban, J. Demilly, N. Dhooge, S. Langlois, P. Le Calvez, M. Potier, F. Relouzat, J. M. Robert and C. Dodan) for help with animal studies, Q. Pascal for performing the necropsies, and F. Ducancel and Y. Gorin for help with the logistics and safety management. C. Petitdemange was supported by the Pasteur-Roux Cantarini fellowship, the Young Investigator Sidaction fellowship and the National Institutes of Health (NIH) grant (R01AI143457). A.O.R. and E.B. received a doctoral fellowship from the Université Paris Cité and the Ministère français de l'Enseignement supérieur, de la Recherche et de l'Innovation. This work was supported by the 'URGENCE COVID-19' fundraising campaign of Institut Pasteur. H.M. received core funding from the Institut Pasteur and the INSERM and was supported by grants from the ANR REACTing Covid19 (20RR028-00), the European Commission Horizon 2020 program (RECoVER project, 101003589), the Institut Pasteur Task Force COVID-19 (2019-NCOV THERAMAB project) and the Fondation de France (00106077). Furthermore, we would like to acknowledge the support from NIH under grants R01DK130472 and R01AI143457 to R.K.R. We gratefully acknowledge the support of Institut Pasteur and Gilead for the purchase of the Fortessa equipment. We are grateful for the support of IDMIT by the French government—Investments for the Future program for infrastructures (PIA) through the ANR-11-INBS-0008 grant as well as from the PIA grant ANR-10-EQPX-02-01 to the FlowCyTech facility at IDMIT. We equally acknowledge the Investments for Future grant ANR-10-INSB-04 to support the UtechS Photonic BioImaging (Imagopole) and C2RT facilities at Institut Pasteur.

## Author contributions

N.H. designed the experiments. N.H., B.J., M.L., C. Petitdemange, C. Planchais, E.B., A.O.R. and P.R. performed the experiments. V.C. and R.L. coordinated the animal studies. B.J. was in charge of the ethical and safety protocols and coordinated the sample transports. N.H., C. Planchais, P.R., E.B. and C. Petitdemange analyzed the data and discussed them with all authors. H.M. and C. Planchais provided the anti-spike Abs and the recombinant spike protein. M.M.T. designed the initial project. M.M.T., F.R. and R.K.R. obtained the funding. N.H. and M.M.T. wrote the manuscript and all co-authors reviewed it.

## Competing interests

The authors declare no competing interests.

## Additional information

**Extended data** is available for this paper at https://doi.org/10.1038/s41590-023-01661-4.

**Correspondence and requests for materials** should be addressed to Nicolas Huot.

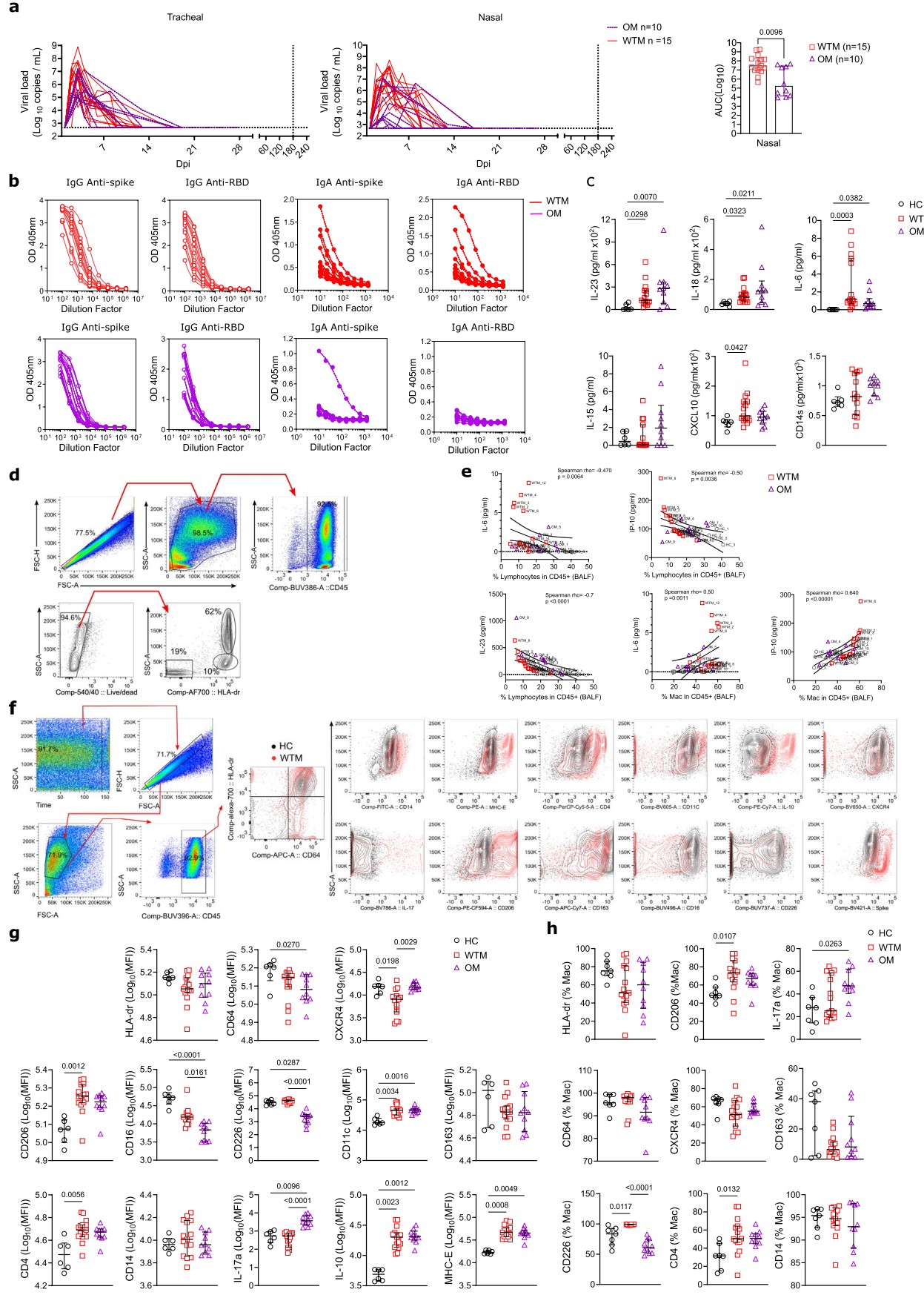

**Extended Data Fig. 1 | See next page for caption.**

**Extended Data Fig. 1 | Viral and immunological parameters in convalescent monkeys. (a)** Viral loads in tracheal and nasal swabs of WTM and OM collected at various time points (2-265 days post-infection (p.i.)) and analyzed for viral genomic RNA (gRNA). The area under the curve (AUC) represents the total gRNA shed between 2 and 21 days p.i. **(b)** Antibody titers (IgG and IgA) measured by ELISA against the entire spike protein or RBD domain in plasma of WTM and OM collected at least 221 days p.i. **(c)** Luminex assay of cytokines (IL-23, -18, -6, and -15) and ELISA assay of cytokine IP-10 and sCD14 in plasma of HC, WTM, and OM collected at least 221 days p.i. **(d)** Flow cytometry analysis gating strategy for BALF Macs at least 221 days p.i., using size and granulometry parameters (FSC-A, FSC-H, SSC-A, SSC-H) and surface expression of CD45 and HLA-DR. **(e)** Spearman correlation between plasmatic cytokines (as in **b**) and the percentage of lymphocytes or Mac among BALF CD45+ cells at least 221 days p.i. **(f)** Gating strategy used to characterize BALF Mac. Representative dot plot for the indicated markers shown for HC and WTM. **(g)** Mean fluorescence intensity (MFI) of various markers in BALF Mac from HC, WTM, and OM at least 221 days p.i. **(h)** Frequency of markers measured in BALF Mac from HC, WTM, and OM at least 221 days p.i. In each panel, individual macaques are represented, and for Spearman correlations, black full lines represent linear regression, while dotted lines represent the confidence interval (95%). Each point is labeled with the corresponding monkey's name. Symbols represent individual macaques, and bars indicate medians. Interquartile ranges are shown. p-Values determined by Kruskal-Wallis test with Dunn's post-test, except for panel **(a)** (left), where a Mann-Whitney test was applied.

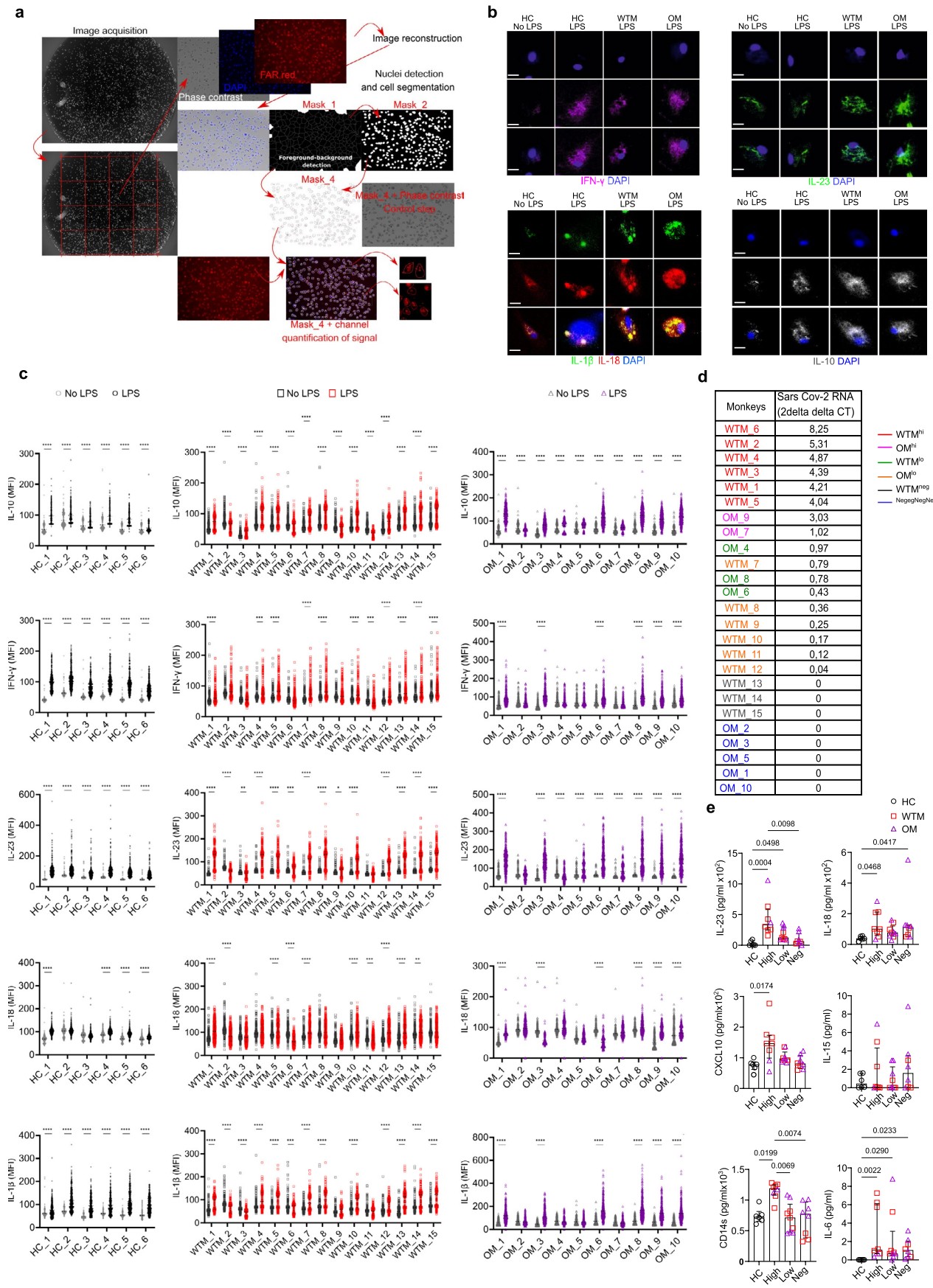

**Extended Data Fig. 2 | See next page for caption.**

**Extended Data Fig. 2 | Mac components in WTM and OM compared to HC.**
**(a)** A 2-D cell segmentation process using image analysis is outlined to quantify cellular and viral components. It involves three stages: (1) Sample preparation and image acquisition, (2) image preprocessing for improved edge detection accuracy, and (3) thresholding-based segmentation using ImageJ. Detailed methodology is in the Methods section. **(b)** High-magnification epifluorescent images show BALF CD64$^+$ Mac isolated from WTM and OM at ≥221 days p.i., as well as HC control images, are shown. These cells were cultured with or without lipopolysaccharide (LPS) for 8 hours and stained for DAPI, IFN-γ, IL-23, IL-1β, IL-18, or IL-10. Scale bars represent 10 μm. **(c)** Cytokine mean fluorescence intensity (MFI) measurements for IL-10, IFN-γ, IL-23, IL-18, and IL-1β expression,

as described in **(a)**, in macrophages isolated from BALF of heathy HC, WTM, and OM at ≥221 days p.i. Macrophages were cultured for 8 hours with or without LPS. Individual cells are displayed in the graph. **(d)** A table summarizes viral load measurements in total BALF cells at ≥221 days p.i. for each animal. **(e)** Cytokine analysis using Luminex and ELISA assays classifies monkeys based on viral RNA levels in total BALF cells at ≥221 days p.i. Each monkey is represented in the graph. Median values with interquartile ranges are shown. Statistical significance was assessed using a 2-way ANOVA test with Šídák's multiple comparisons test correction in panel **c** (***=p < 0.0001, **=p < 0.001, *=p < 0.01) and Kruskal-Wallis test with Dunn's post-test in panel **d**.

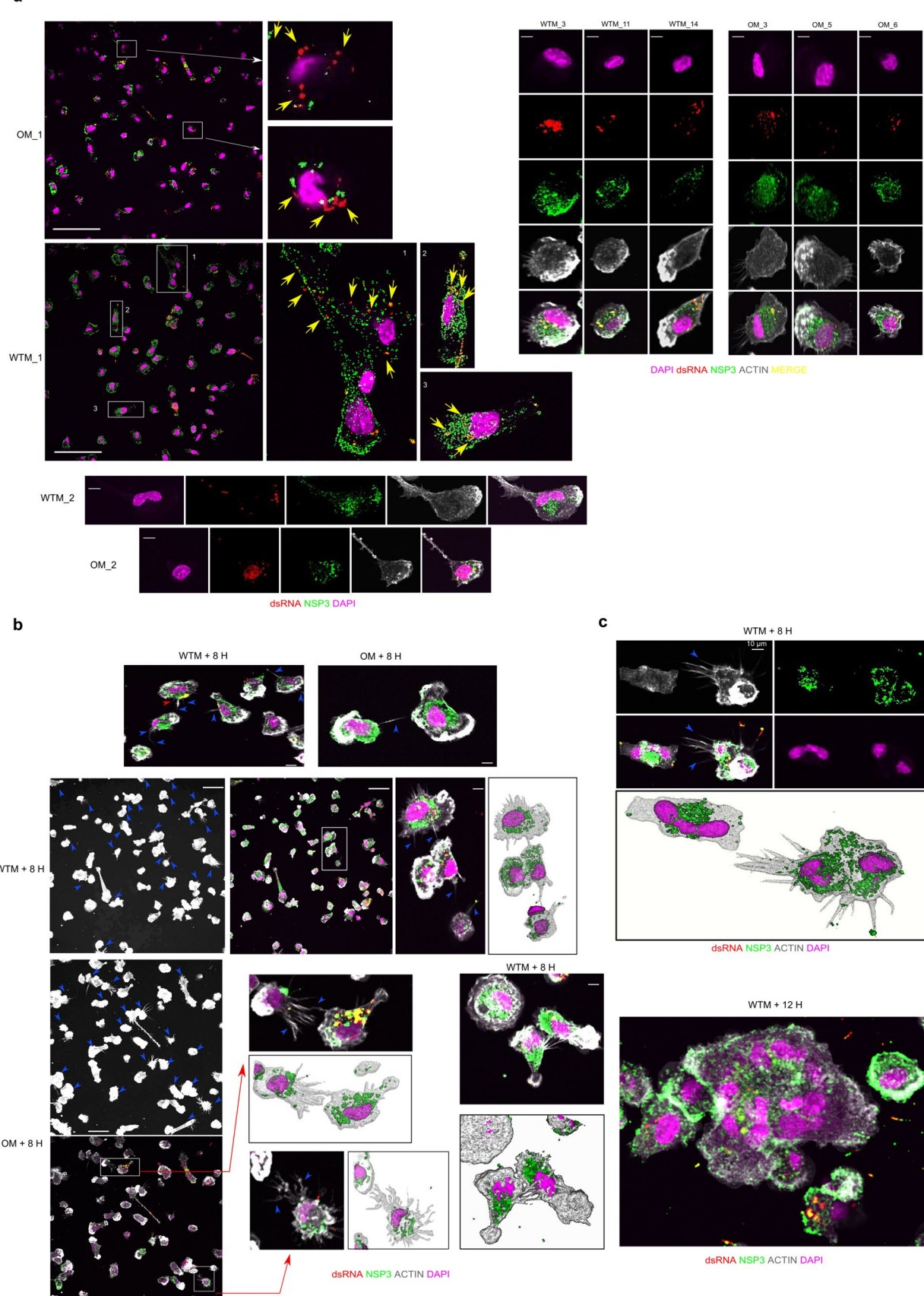

**Extended Data Fig. 3 | See next page for caption.**

**Extended Data Fig. 3 | Viral compound detection in cultured BALF macrophages. (a)** Confocal microscopy images show the intracellular localization and accumulation of SARS-CoV-2 nonstructural protein 3 (NSP3) and double-stranded RNA (dsRNA) in BALF CD64⁺ Mac isolated from WTM and OM at ≥221 days p.i. After 8 hours of culture, cells were fixed with 4% formaldehyde. Immunostaining included DAPI (nucleus; purple), anti-dsRNA (red), and anti-NSP3 (green). Yellow arrows highlight dsRNA within the macrophages. **(b)** Similar to **(a)**, these confocal microscopy images depict the intracellular localization of NSP3 and dsRNA in BALF CD64⁺ Macs, with the addition of Phalloidin (white) staining to highlight protrusions within the macrophages. Blue arrows indicate the presence of protrusions. Increased threshold on fields displaying only Actin was used to facilitate visualization of pseudopod- and TNT-like structures. **(c)** Confocal microscopy images as in **(b)** show syncytia observed in MAC culture. Scale bars in **(a)** = 45 µm, Scale bars in **(b)** and **(c)** = 35 µm. Some images use 3D rendering techniques to highlight the three-dimensional aspect of the visualization.

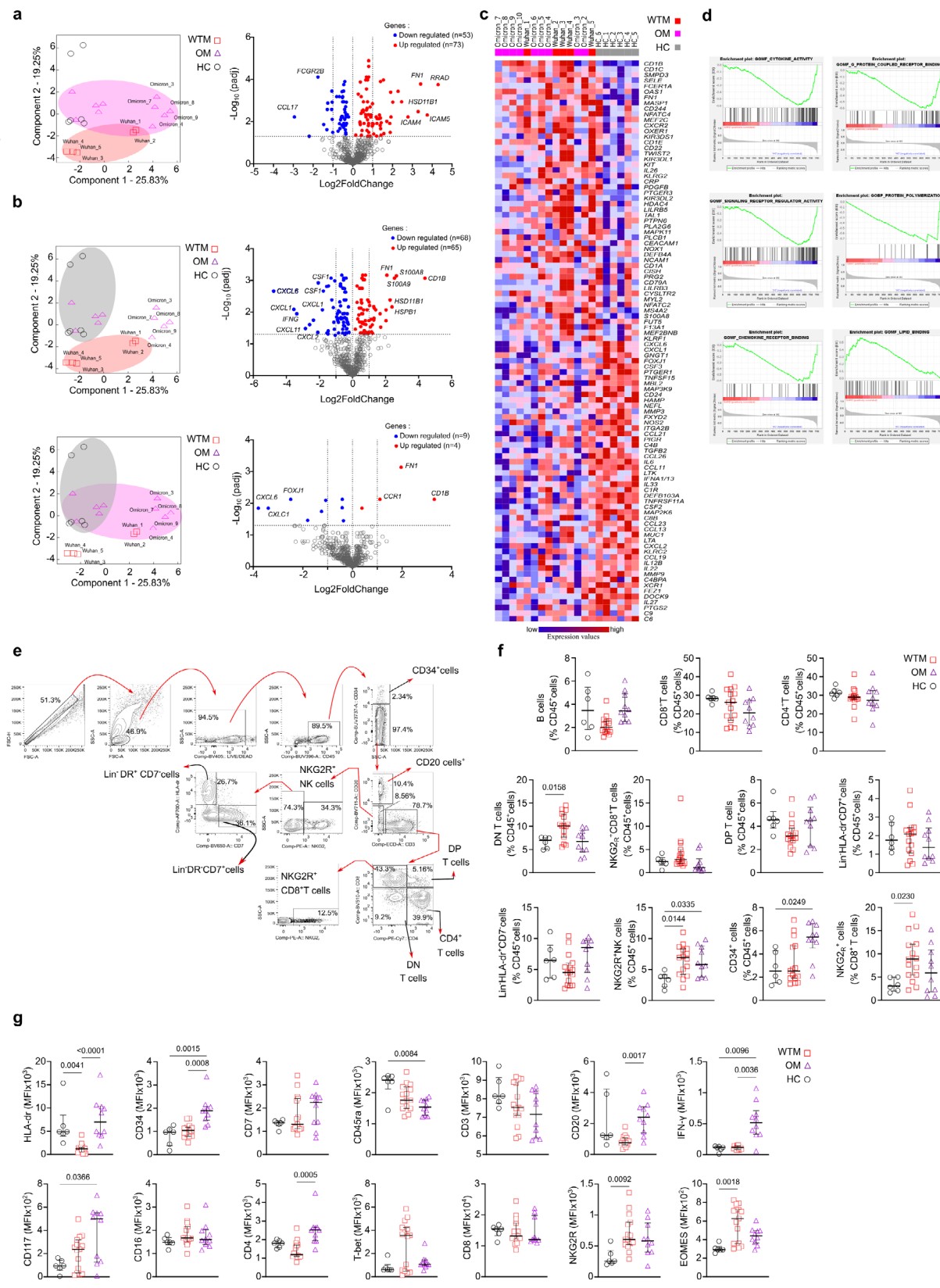

**Extended Data Fig. 4 | See next page for caption.**

**Extended Data Fig. 4 | In-vitro transcriptome analyses of BALF macrophages.**
CD64$^+$ macrophages were isolated from BALF of 5 HC, 5 randomly selected WTM, and 10 OM at least 221 days p.i. After 8 hours of culture, they were profiled using the nanoString nCounter System. **(a)** Principal Component Analysis (PCA) shows clustering patterns of convalescent monkeys. OM samples are represented by purple triangles and WTM samples by red squares. Volcano plots highlight genes significantly up-regulated and down-regulated between WTM and OM.
**(b)** Similar to **(a)**, volcano plots highlight genes significantly up-regulated and down-regulated between WTM and HC (up) or between OM and HC (down).
**(c)** Heat map displays the top 50 genes resulting from Gene Set Enrichment Analysis (GSEA) conducted on the Hallmark gene sets obtained in **(a)**. **(d)** Enrichment plots show data sets enriched in GSEA Hallmark analysis of CD64$^+$ Mac as in (a), displaying the running Enrichment Score (ES) and positions of gene set members on the rank-ordered list. **(e)** Gating strategy defines different lymphocyte populations in BALF of HC, WTM, and OM at least 221 days p.i. Populations are defined based on specific marker expression. **(f)** Comparison of various lymphocyte populations in BALF of HC, WTM, and OM. **(g)** Comparison of marker expression on BALF CD45$^+$ lymphocytes as described in **(e)**. In each graph, individual symbols represent individual monkeys, bars indicate medians, and interquartile ranges are shown. p-Values were determined using a Kruskal-Wallis test with Dunn's post-test in **(f)** and **(g)**.

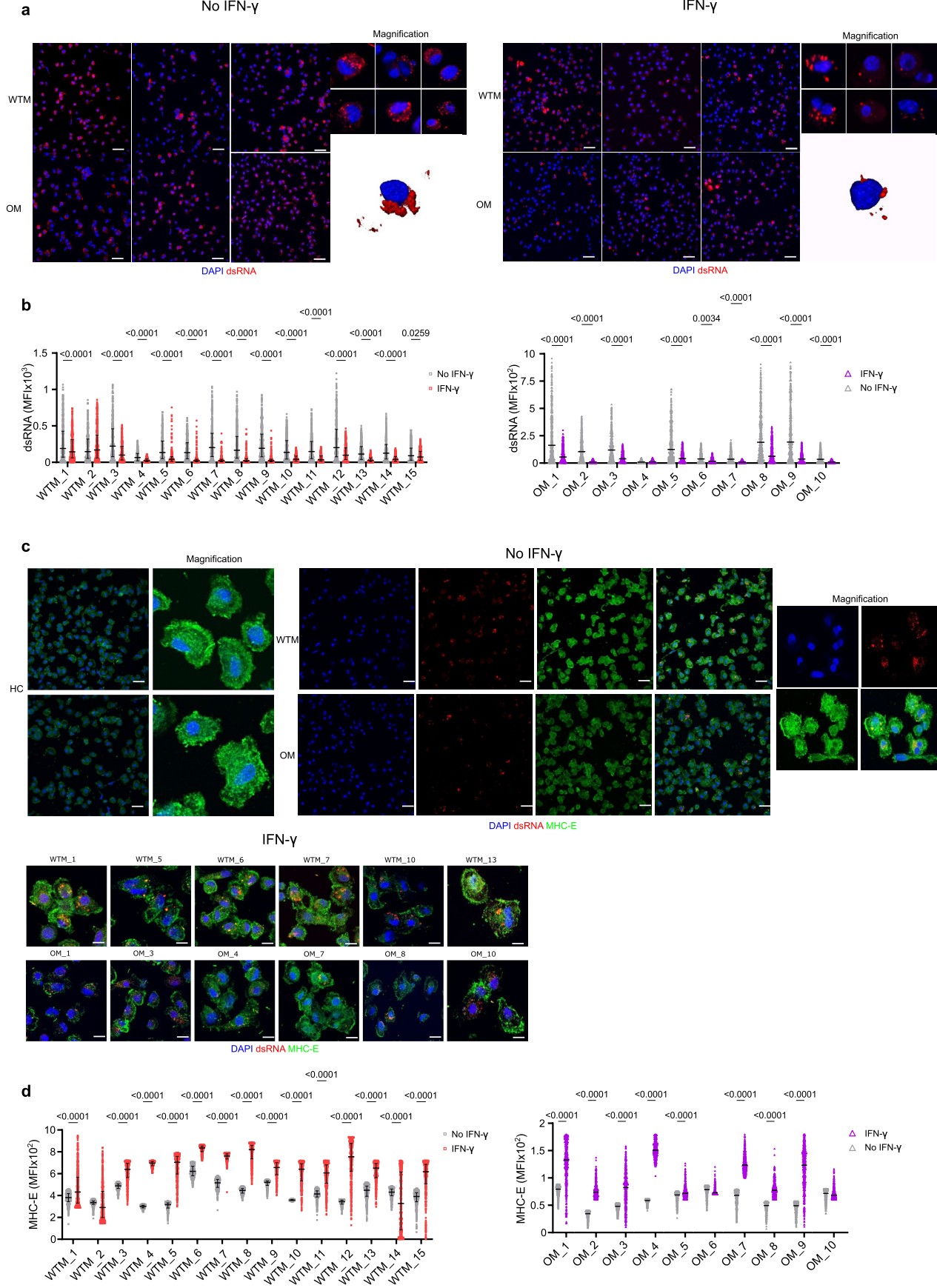

**Extended Data Fig. 5 | See next page for caption.**

**Extended Data Fig. 5 | Impact of IFN-γ on SARS-CoV-2 replication in BALF macrophages. (a)** Confocal microscopy images show the intracellular accumulation of SARS-CoV-2 dsRNA in BALF CD64⁺ Mac from WTM and OM at ≥221 days p.i., cultured with or without IFN-γ. DAPI (nucleus) is in purple, and anti-dsRNA staining (SARS-CoV-2 dsRNA) is in red. **(b)** Measurement of SARS-CoV-2 dsRNA MFI in CD64⁺ Macs isolated from BALF of WTM, and OM at ≥221 days p.i., cultured with or without IFN-γ. **(c)** Confocal microscopy images depict MHC-E localization in BALF CD64⁺ Macs from WTM and OM at ≥221 days p.i., cultured with or without IFN-γ stimulation. DAPI (nucleus) is in blue, anti-dsRNA staining (SARS-CoV-2 dsRNA) is in red, and anti-MHC-E staining is in green. **(d)** Measurement of MHC-E MFI in CD64⁺ Macs isolated from BALF of WTM, and OM at ≥221 days p.i., cultured with or without IFN-γ. In each graph, median and interquartile range are shown. p-Values were determined using a 2-way ANOVA test with Šídák's multiple comparisons test correction.

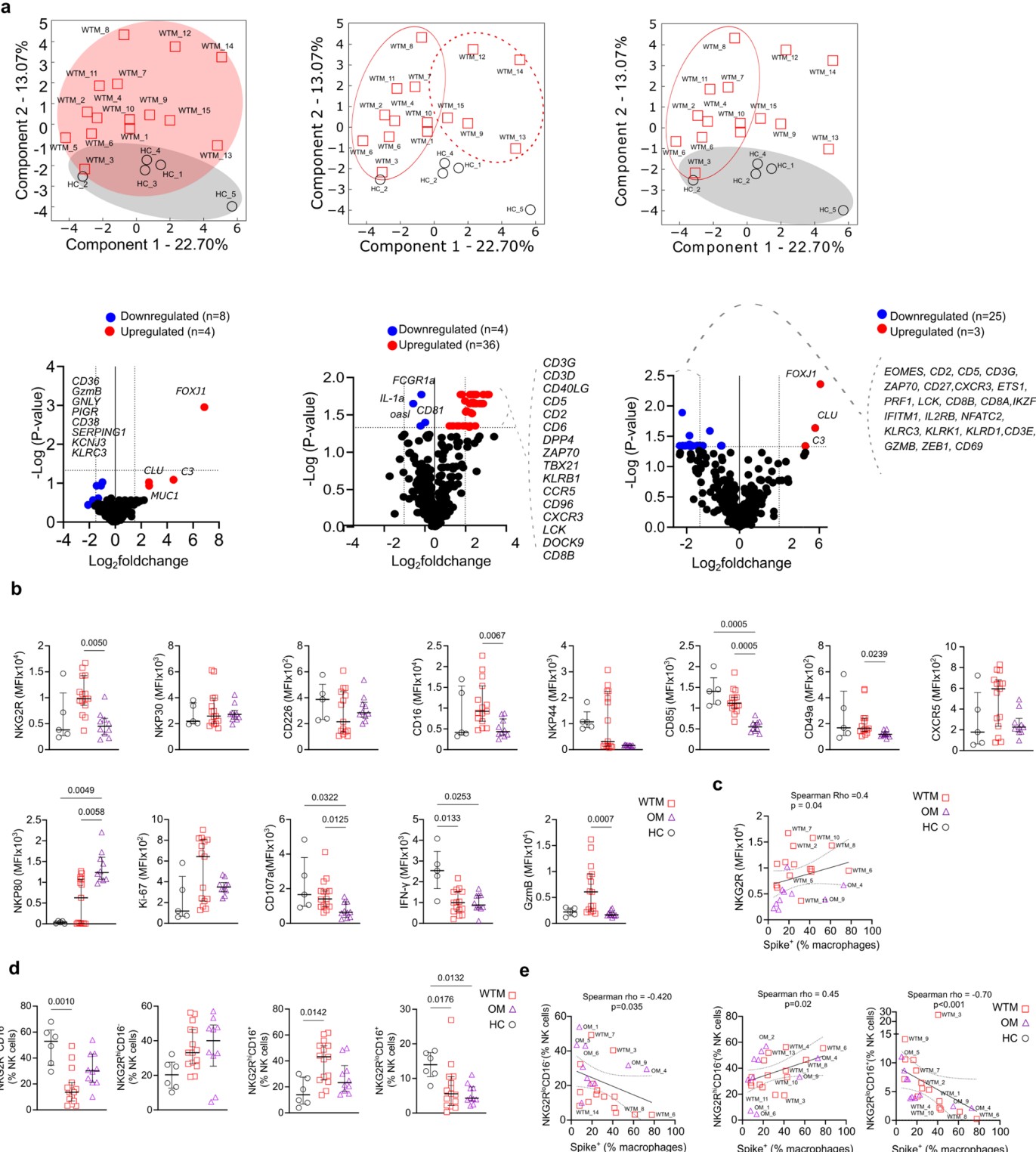

**Extended Data Fig. 6 | Transcriptome analysis of BALF NKG2R⁺ NK cells.**
**(a)** Top: PCA shows the clustering pattern of WTM and HC samples. Red circles represent WTM, and gray circles represent HC. Each PCA plot indicates the specific comparison performed. Bottom: Volcano plots highlight genes significantly up-regulated and down-regulated in various comparisons, using p-values instead of Padj. Horizontal lines represent p-value = 0.05, and vertical lines indicate log2 fold change > 1 or < -1. **(b)** Comparison of MFI for various markers on BALF NKG2R⁺ NK cells isolated from HC, WTM, and OM at least 221 days p.i. **(c)** Spearman correlation analysis between NKG2R MFI on BALF NKG2R⁺ NK cells from WTM and OM isolated at least 221 days p.i. and Spike⁺

Mac measured as in Fig. 2. **(d)** Comparison of NKG2R^lo CD16⁻, NKG2R^hi CD16⁻, NKG2R^hi CD16⁺, and NKG2R^lo CD16⁺ populations among bulk NK cells, measured on BALF NKG2R⁺ NK cells from HC, WTM, and OM isolated at least 221 days p.i. **(e)** Spearman correlation between the frequency of NKG2R^hi CD16⁻, NKG2R^hi CD16⁺, and NKG2R^lo CD16⁺ populations among bulk NK cells (as in **(d)**) and Spike⁺ Mac measured as in Fig. 2. For all Spearman correlations, the black solid line represents linear regression, and dotted lines indicate the confidence interval (95%). Symbols represent individual macaques, bars indicate medians, and interquartile ranges are shown. p-Values were determined by Kruskal-Wallis test with Dunn's post-test.

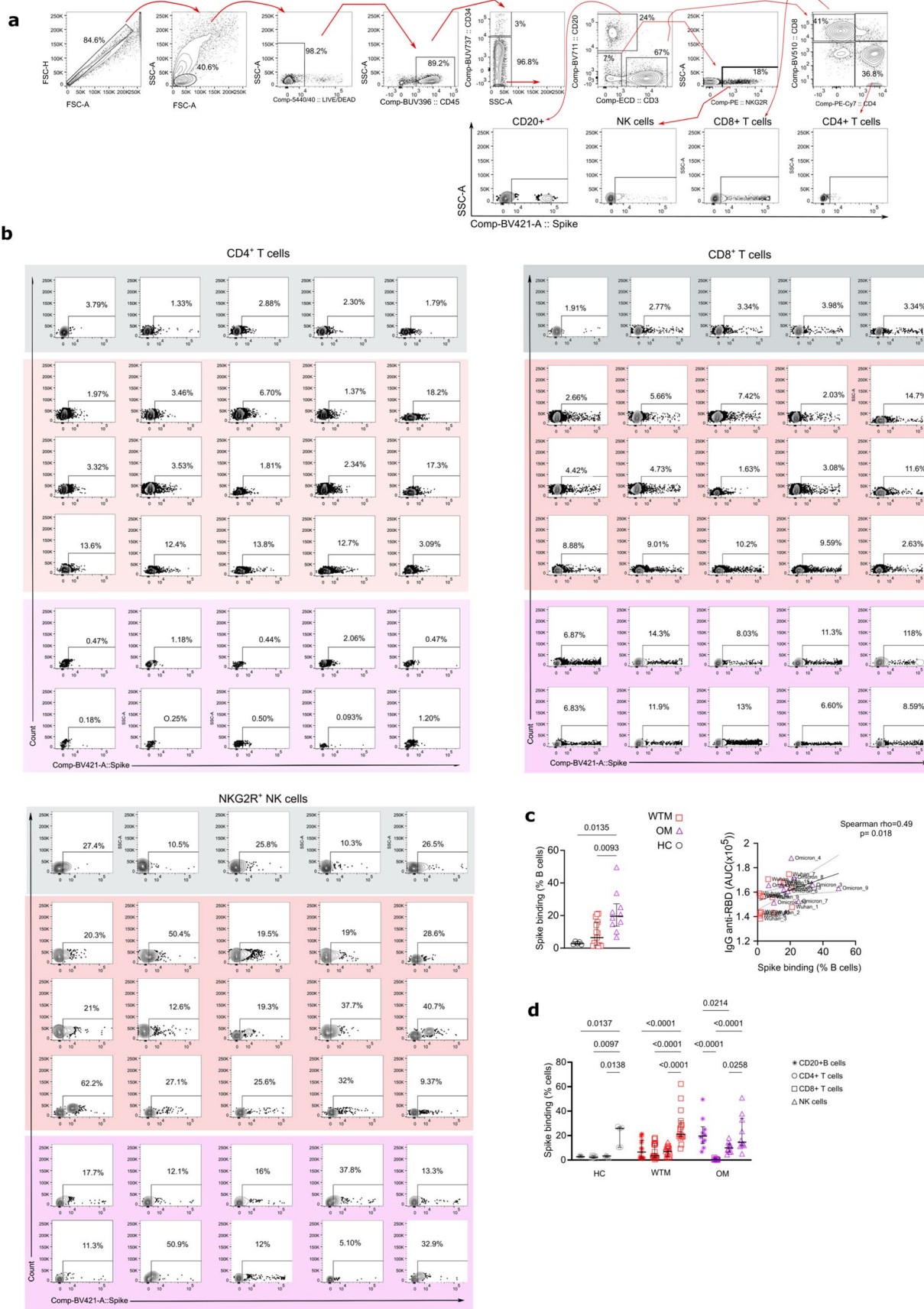

**Extended Data Fig. 7 | See next page for caption.**

**Extended Data Fig. 7 | SARS-CoV-2 spike protein binding on lymphocytes.**
**(a)** Representative flow cytometry plots showing the gating strategy for trimeric spike surface staining on total BALF cells from HC, WTM, and OM at ≥221 days p.i. Populations are defined based on specific marker expression. **(b)** Dot plot illustrating the percentage of surface trimeric spike staining on CD4+ T cells, CD8+ T cells, and NKG2R+ NK cells from HC, WTM, and OM. Dot plots are color-coded for clarity. **(c)** Left: graph displaying the percentage of CD20+ B cells positive for trimeric spike surface staining in total BALF cells from HC, WTM, and OM at ≥221 Days p.i. Right: positive Spearman correlation between the percentage of cells stained with trimeric spike among BALF CD20+ B cells and the amount of IgG anti-RBD measured in the plasma as in Fig. 1. **(d)** Graph showing the percentage of CD4+ T cells, CD8+ T cells, and NK cells positive for trimeric spike surface staining in BALF cells from HC, WTM, and OM at ≥221 Days p.i. In Spearman correlation, the black solid line represents linear regression, and the dotted line indicates the confidence interval (95%). Each point is labeled with the corresponding monkey's name. In each graph, dots, squares, or triangles represent individual monkeys, bars indicate medians, and interquartile ranges are shown. p-Values were determined using a Kruskal-Wallis test with Dunn's post-test.

**a**

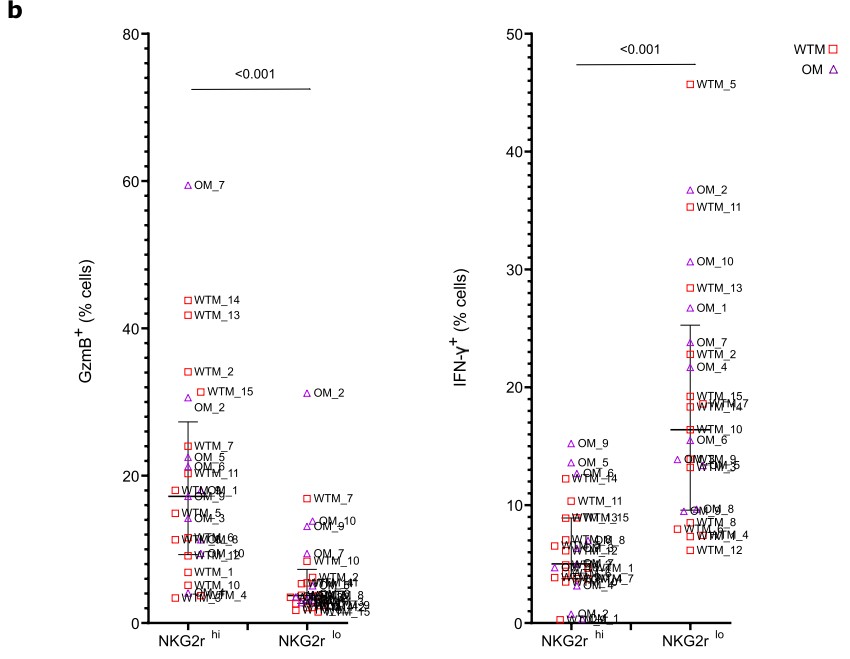

**Extended Data Fig. 8 | See next page for caption.**

**Extended Data Fig. 8 | NK cell responses after exposure to SARS-CoV-2 spike protein. (a)** Representative dot plot illustrating the percentage of intracellular GzmB or IFN-γ in NK cells obtained from BALF of HC, WTM, and OM at ≥ 221 days p.i following 24 hours of culture exposed or non-exposed to trimeric spike protein. The red and purple light squares around the dot plot delineate the WTM (n = 10) and OM (n = 10) convalescent monkey groups, respectively.

**(b)** Comparative analysis of intracellular GzmB or IFN-γ between NKG2R$^{hi}$ and NKG2R$^{lo}$ NK cells, following 24 hours of exposure to trimeric spike protein. Each symbol represents an individual monkey, and the median along with the interquartile range is depicted. The p-value was determined using the Wilcoxon matched-pairs signed rank test.

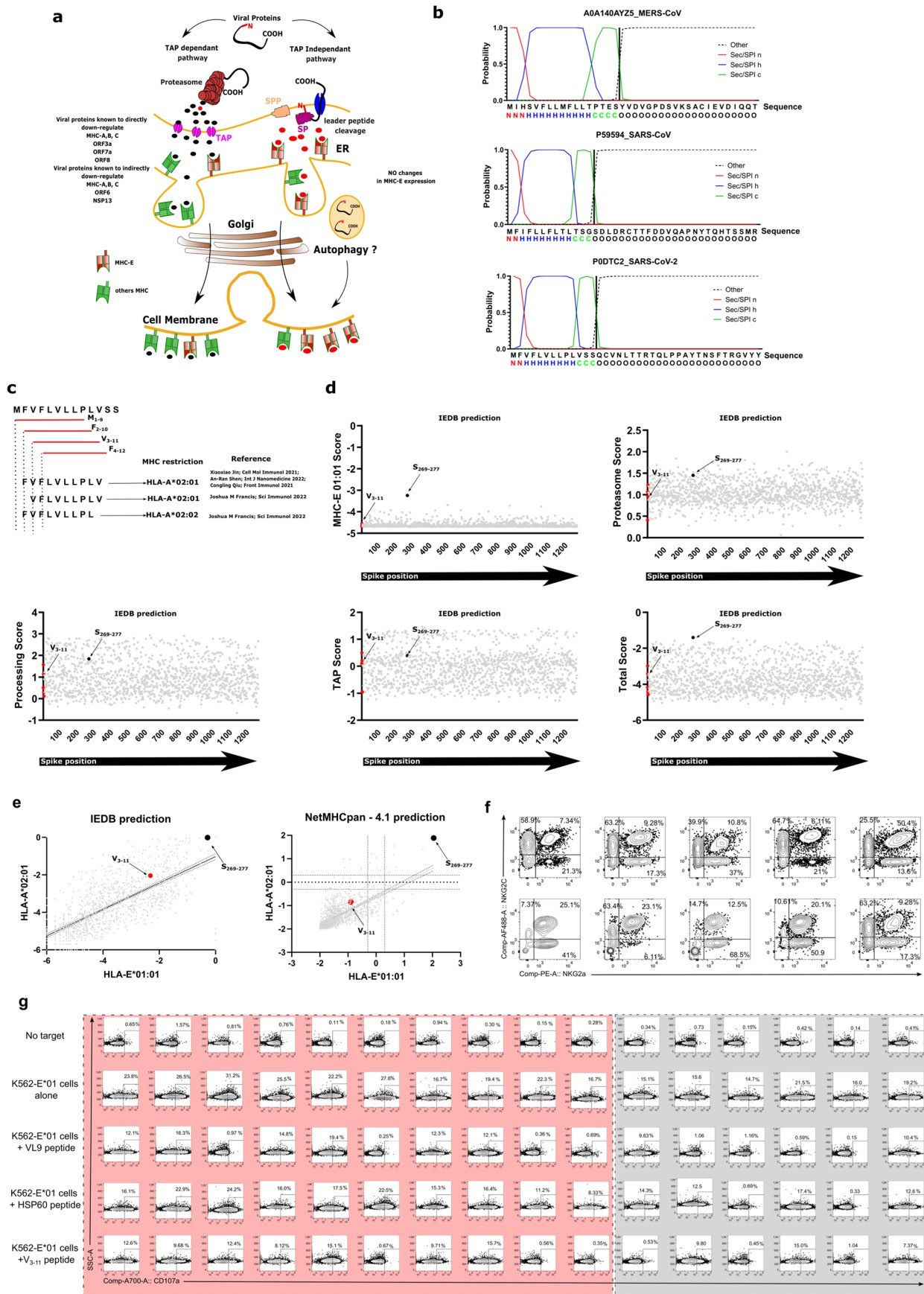

**Extended Data Fig. 9 | See next page for caption.**

**Extended Data Fig. 9 | Identifying HLA-E Binding Potential in SARS-CoV-2 Signal Peptides. (a)** Literature-based scheme summarizing peptide processing via TAP-dependent and TAP-independent pathways. **(b)** In silico analysis predicting signal peptide presence in spike proteins and their cleavage sites, with SignalP 6.0 software suggesting cleavage by Signal peptidase (SP) and Signal peptide peptidase (SPP). Signal peptides comprise hydrophobic core (h region), polar C-terminal end (c region), and polar N-terminal region (n region). **(c)** Schematic displaying four overlapping nonamers in the 5' region of SARS-CoV-2 signal peptide (P0DTC2), with V3-11 resembling the MHC-E binding motif. MHC restrictions and references provided. **(d)** In silico epitope predictions for HLA-E binding using the IEDB analysis resource, covering the entire spike sequence. Graphs exhibit HLA-E*01:01 binding scores, proteasomal cleavage, transport scores, and total scores combining HLA-E binding and processing. Red dots represent spike SP peptides from **(c)**, and S269–277 corresponds to a previously described peptide. **(e)** Comparison of binding probabilities to HLA-E and HLA-A using NetMHCpan - 4.1, analyzing the entire SARS-CoV-2 spike protein. Multiple nonamers represented as black dots, with dot positions indicating predicted HLA-E binding capacity. Percentile ranks for HLA-E *01:01 (y-axis) and HLA-E* 02:01 (x-axis) are provided. **(f)** Representative dot plot showcasing NKG2A[+] and NKG2C[+] human NK cell responses. **(g)** Dot plots displaying the percentage of CD107a[+] NK cells isolated from HC and WTM at ≥ 221 days p.i. after co-culture with K562-E*01 cells loaded with peptides (VL9, HSP60, or V[3-11]) or no peptides. Light squares indicate wildtype infected convalescent monkeys, while light gray squares represent HC.

**Extended Data Table 1 | Baseline characteristics of Cynomolgus macaque monkeys involved in this study**

| Name | Species | Sex | Birth Date | MHC Haplotype 1 | MHC Haplotype 2 | Strain used for infection | SARS-CoV-2 infection date | Age of the animal on the date of the 1st sample (years) | First sampling time after inoculation for this study (days) | Treatment during acute SARS-COV-2 infection | Reused animal (according to 3 R rule) | Other previous interventions before this study | Relapse time between previous intervention and inclusion in this study (months) |
|---|---|---|---|---|---|---|---|---|---|---|---|---|---|
| HC_1 | Cyno | F | 14/03/2018 | ND | ND | NA | NA | 3.3 | NA | NO | NO | NONE | NA |
| HC_2 | Cyno | F | 22/06/2018 | ND | ND | NA | NA | 3.07 | NA | NO | NO | NONE | NA |
| HC_3 | Cyno | M | 28/09/2018 | ND | ND | NA | NA | 2.8 | NA | NO | NO | NONE | NA |
| HC_4 | Cyno | F | 24/07/2018 | ND | ND | NA | NA | 2.9 | NA | NO | NO | NONE | NA |
| HC_5 | Cyno | M | 15/03/2018 | ND | ND | NA | NA | 3.3 | NA | NO | NO | NONE | NA |
| HC_6 | Cyno | F | 18/06/2018 | ND | ND | NA | NA | 3.0 | NA | NO | NO | NONE | NA |
| Wuhan_2 | Cyno | F | 20/12/2016 | H1 | Rec H4 H1 | hCoV-19/France/IDF0372/2020 | 25/03/2020 | 4.5 | 479 | NO | YES | Non-SARS-CoV-2 related immunisation | 7 |
| Wuhan_4 | Cyno | F | 06/01/2017 | Rec H3 H5 | Rec H2 H1 | hCoV-19/France/IDF0372/2020 | 10/04/2020 | 4.4 | 463 | NO | YES | Non-SARS-CoV-2 related immunisation | 8 |
| Wuhan_5 | Cyno | M | 14/02/2017 | H4 | Rec H4 / H5 | hCoV-19/France/IDF0372/2020 | 10/04/2020 | 4.3 | 463 | NO | YES | Non-SARS-CoV-2 related immunisation | 7 |
| Wuhan_7 | Cyno | M | 12/02/2017 | H3 | Rec H1 H5 | hCoV-19/France/IDF0372/2020 | 25/03/2020 | 4.3 | 479 | YES (Hydroxychloroquine) | YES | Non-SARS-CoV-2 related immunisation | 8 |
| Wuhan_15 | Cyno | M | 13/02/2017 | Rec H3 H5 | Rec H2 H1 | hCoV-19/France/IDF0372/2020 | 15/04/2020 | 4.2 | 461 | YES (Hydroxychloroquine) | YES | Non-SARS-CoV-2 related immunisation | 8 |
| Wuhan_1 | Cyno | M | 14/02/2017 | H1 | Rec H4 (H2) H1 | hCoV-19/France/IDF0372/2020 | 15/04/2020 | 4.2 | 461 | YES (Hydroxychloroquine) | YES | Non-SARS-CoV-2 related immunisation | 7 |
| Wuhan_3 | Cyno | F | 22/02/2017 | H4 | H2 | hCoV-19/France/IDF0372/2020 | 15/04/2020 | 4.3 | 461 | YES (Hydroxychloroquine) | YES | Non-SARS-CoV-2 related immunisation | 7 |
| Wuhan_10 | Cyno | M | 26/02/2017 | H3 | Rec H4 H1 | hCoV-19/France/IDF0372/2020 | 15/04/2020 | 4.2 | 461 | YES (Hydroxychloroquine) | YES | Non-SARS-CoV-2 related immunisation | 8 |
| Wuhan_6 | Cyno | M | 04/03/2017 | Rec H1 / H2 | H7 | hCoV-19/France/IDF0372/2020 | 15/04/2020 | 4.1 | 461 | NO | NO | Non-SARS-CoV-2 related immunisation | 7 |
| Wuhan_9 | Cyno | F | 13/03/2017 | H3 | H5 | hCoV-19/France/IDF0372/2020 | 27/10/2020 | 4.1 | 221 | NO | NO | NONE | NA |
| Wuhan_8 | Cyno | M | 03/03/2017 | H2 | Rec H3 H2 | hCoV-19/France/IDF0372/2020 | 27/10/2020 | 4.3 | 221 | NO | NO | NONE | NA |
| Wuhan_11 | Cyno | M | 16/03/2017 | ND | ND | hCoV-19/France/IDF0372/2020 | 27/10/2020 | 4.2 | 233 | NO | NO | NONE | NA |
| Wuhan_12 | Cyno | F | 03/04/2017 | H3 | Rec H7 H4 | hCoV-19/France/IDF0372/2020 | 27/10/2020 | 4.1 | 190 | NO | NO | NONE | NA |
| Wuhan_13 | Cyno | F | 02/01/2016 | H1 | Rec H4 H3 | hCoV-19/France/IDF0372/2020 | 27/10/2020 | 5.3 | 202 | NO | NO | NONE | NA |
| Wuhan_14 | Cyno | F | 22/08/2017 | H1 | H1 | hCoV-19/France/IDF0372/2020 | 27/10/2020 | 3.7 | 233 | NO | NO | NONE | NA |
| Omicron_1 | Cyno | F | 20/03/2017 | H6 | H2 | SARS-CoV O□□□□□□□ | □□□□□□□□ | □□□ | □□□ | □□ | □□ | □□□□ | □□ |
| □□□□□□□□ □ O□□□ | □ | □ | □□□□□□□□ □□ O□ | □□□□□□O□ O□ | O□□ | □□□□□□O□O□□□□□□□ | □□□□□□□□ □ | □□□ | □□□ | □□ | □□ | □□□□ | □□ |
| □□□□□□□□ □ O□□□ | □ | □ | □□□□□□□□ □□ O□□ | O□□ | □□□□□□O□O□□□□□□□ | □□□□□□□□ □ | □□□ | □□□ | □□ | □□ | □□□□ | □□ |
| □□□□□□□ □ O□□□ | □ | □ | □□□□□□□□ □□ | □□□ | □□□□□O□O□□□□□□□□ | □□□□□□□□ □ | □□□ | □□□ | □□ | □□ | □□□□ | □□ |
| □□□□□□□□ □ O□□□ | □ | □ | □□□□□□□□ □□ | □□□ | □□□□□□O□O□□□□□□□ | □□□□□□□□ □ | □□□ | □□□ | □□ | □□ | □□□□ | □□ |
| □□□□□□□□ □ O□□□ | □ | □ | □□□□□□□□ □□□O□□ □O□ | O□□ O□ | □□□□□O□O□□□□□□□□ | □□□□□□□□ □ | □□□ | □□□ | □□ | □□ | □□□□ | □□ |
| □□□□□□□□ □ O□□□ | □ | □ | □□□□□□□□ □□ O□ | □□□□□O□□ | □□□□□O□O□□□□□□□□ | □□□□□□□□ □ | □□□ | □□□ | □□ | □□ | □□□□ | □□ |
| □□□□□□□□ □ O□□□ | □ | □ | □□□□□□□□ □□ O□□ | O□□ | □□□□□O□O□□□□□□□□ | □□□□□□□□ □ | □□□ | □□□ | □□ | □□ | □□□□ | □□ |
| □□□□□□□ □ O□□□ | □ | □ | □□□□□□□□ □□ O□□ | O□□ | □□□□□O□O□□□□□□□□ | □□□□□□□□ □ | □□□ | □□□ | □□ | □□ | □□□□ | □□ |

For all the animals the date of first sampling used to perform the analysis is indicated. The second sampling occurred 2-8 weeks later depending on the monkeys. NA means not applicable. HC are highlighted in gray, WTM in blue and OM in red.

# Reporting Summary

## Statistics

For all statistical analyses, confirm that the following items are present in the figure legend, table legend, main text, or Methods section.

| n/a | Confirmed | |
|---|---|---|
| ☐ | ☒ | The exact sample size (*n*) for each experimental group/condition, given as a discrete number and unit of measurement |
| ☐ | ☒ | A statement on whether measurements were taken from distinct samples or whether the same sample was measured repeatedly |
| ☐ | ☒ | The statistical test(s) used AND whether they are one- or two-sided<br>*Only common tests should be described solely by name; describe more complex techniques in the Methods section.* |
| ☐ | ☒ | A description of all covariates tested |
| ☐ | ☒ | A description of any assumptions or corrections, such as tests of normality and adjustment for multiple comparisons |
| ☐ | ☒ | A full description of the statistical parameters including central tendency (e.g. means) or other basic estimates (e.g. regression coefficient) AND variation (e.g. standard deviation) or associated estimates of uncertainty (e.g. confidence intervals) |
| ☐ | ☒ | For null hypothesis testing, the test statistic (e.g. *F*, *t*, *r*) with confidence intervals, effect sizes, degrees of freedom and *P* value noted<br>*Give P values as exact values whenever suitable.* |
| ☒ | ☐ | For Bayesian analysis, information on the choice of priors and Markov chain Monte Carlo settings |
| ☒ | ☐ | For hierarchical and complex designs, identification of the appropriate level for tests and full reporting of outcomes |
| ☐ | ☒ | Estimates of effect sizes (e.g. Cohen's *d*, Pearson's *r*), indicating how they were calculated |

*Our web collection on statistics for biologists contains articles on many of the points above.*

## Software and code

Policy information about availability of computer code

| Data collection | Flow cytometry analysis was performed using a Fortessa (BD Biosciences) equipped with DIVA software (version 8.0). Immunofluorescence analysis was carried out using either a CELENA® X High Content Imaging System (LOGOS BIOSYSTEMS, South Korea) with the CELENA® X Explorer (version 1.6.0) or a spinning-disk Ti2E Microscope (Nikon - Yokogawa CSU W1). Thermal cycling was conducted in a 7500 real-time PCR system (Applied Biosystems) equipped with Windows 10 and 7500 Fast software version 2.3. |
|---|---|
| Data analysis | The experimental group size estimation was performed using G Power software version 3.1.9.7, and fluorescence quantification and cell segmentation were carried out using the Fiji software (version 1.53 with Java 1.8.0_322). In the quest to identify SARS-CoV-2 peptides with the potential for efficient binding to MHC-E, IEBD prediction tools (version 2.27) were employed, and Signalp software (version 6.0) was used for further analysis. Cytometry analysis was conducted with FlowJo software (version 10.4.2, Tree Star, Ashland, OR). Subsequently, data analysis and statistical tests were performed using GraphPad Prism 7 (GraphPad Software, version 10.0.2, San Diego, CA). |

For manuscripts utilizing custom algorithms or software that are central to the research but not yet described in published literature, software must be made available to editors and reviewers. We strongly encourage code deposition in a community repository (e.g. GitHub). See the Nature Portfolio guidelines for submitting code & software for further information.

## Data

Policy information about [availability of data](availability of data)
All manuscripts must include a [data availability statement](data availability statement). This statement should provide the following information, where applicable:
- Accession codes, unique identifiers, or web links for publicly available datasets
- A description of any restrictions on data availability
- For clinical datasets or third party data, please ensure that the statement adheres to our [policy](policy)

All the data that support the study's findings are included either within the article itself or are provided in a format that readers can access directly from the article. The authors provide raw data files associated with the study. These files likely contain detailed data points and information related to the research. all figures presented in the manuscript are linked to the corresponding raw data files.
All omics data are deposited in public repositories and access codes specified in the paper.

# Field-specific reporting

Please select the one below that is the best fit for your research. If you are not sure, read the appropriate sections before making your selection.

☒ Life sciences ☐ Behavioural & social sciences ☐ Ecological, evolutionary & environmental sciences

For a reference copy of the document with all sections, see [nature.com/documents/nr-reporting-summary-flat.pdf](nature.com/documents/nr-reporting-summary-flat.pdf)

# Life sciences study design

All studies must disclose on these points even when the disclosure is negative.

| | |
|---|---|
| Sample size | We followed the "3Rs" principle for animal research, aiming to minimize the use of experimental subjects. Animals were selected to closely match in terms of age, weight, and genotype. Sample size calculations were performed using power analysis, which accounts for effect size, standard deviation, type 1 error, and 80% power in a two-sample t-test with a 5% significance level (two-sided test). We used G power software version 3.1.9.7. for these calculations, with D determined as (difference in means)/(standard deviation), estimated from our preliminary results. We considered a maximum D value of 1.8 based on previous studies for sample size calculation. |
| Data exclusions | No data were excluded from the analyses. |
| Replication | Some assays were performed as replicates: viral quantifications and competitive analysis for binding of the peptides to MHC. The latter comprised three replicates (three independent experiments for each condition). We did not observe any major variation with all the replicates. All results were included in the manuscript. For the other assays, we only used biological replica (ie distinct animals). No other attempts were made at replication due to the limitation in volume of samples, and also duration of the in vivo study (2 years), substantial costs associated with animal acquisition, per diem charges, and surgical costs. |
| Randomization | Sample collection and analyses were performed in random order. |
| Blinding | The investigators were not blinded while the animal handlers were blinded to group allocation. |

# Reporting for specific materials, systems and methods

We require information from authors about some types of materials, experimental systems and methods used in many studies. Here, indicate whether each material, system or method listed is relevant to your study. If you are not sure if a list item applies to your research, read the appropriate section before selecting a response.

## Materials & experimental systems

| n/a | Involved in the study |
|---|---|
| ☐ | ☒ Antibodies |
| ☐ | ☒ Eukaryotic cell lines |
| ☒ | ☐ Palaeontology and archaeology |
| ☐ | ☒ Animals and other organisms |
| ☐ | ☒ Human research participants |
| ☒ | ☐ Clinical data |
| ☒ | ☐ Dual use research of concern |

## Methods

| n/a | Involved in the study |
|---|---|
| ☒ | ☐ ChIP-seq |
| ☐ | ☒ Flow cytometry |
| ☒ | ☐ MRI-based neuroimaging |

## Antibodies

| | |
|---|---|
| Antibodies used | These antibodies, along with their respective characteristics, were crucial for ensuring the accuracy and reliability of our research findings. |

CD45, labeled with BUV395 and provided by BD Biosciences (Clone: D058-1283, Isotype: IgG1, κ, Reference: 564099), was used at a dilution of 1/30. Control conditions included Isotype + FMO and Providers + NHP reagent. CD4, labeled with PerCp-Cy5 and sourced from BD Biosciences (Clone: L200, Isotype: IgG1, κ, Reference: 552838), was utilized at a dilution of 1/15. Control conditions comprised Isotype + FMO and Providers + NHP reagent.

CD336 (NKp44), labeled with PC7 and obtained from Miltenyi Biotec (Clone: REA1163, Isotype: IgG1, Reference: 130-120-359), was used at a dilution of 1/20. Control conditions included Isotype + FMO and Providers + NHP reagent.CD337 (NKp30), labeled with APC and sourced from Miltenyi Biotec (Clone: REA823, Isotype: IgG1, Reference: 130-112-431), was employed at a dilution of 1/20. Control conditions comprised Isotype + FMO and Providers + NHP reagent.CD107a, labeled with A700 and provided by BD Biosciences (Clone: H4A3, Isotype: IgG1, κ, Reference: 561340), was used at a dilution of 1/10. Control conditions included Isotype + FMO and Providers + NHP reagent. NKp80, labeled with APC Vio® 770 and sourced from Miltenyi Biotec (Clone: REA845, Isotype: IgG1, Reference: 130-112-593), was utilized at a dilution of 1/20. Control conditions comprised Isotype + FMO and Providers + NHP reagent. CD226, labeled with BV605 and obtained from BD Biosciences (Clone: DX11, Isotype: IgG1, κ, Reference: 742495), was used at a dilution of 1/15. Control conditions included Isotype + FMO and Providers + NHP reagent. CD20, labeled with BV711 and provided by BD Biosciences (Clone: 2H7, Isotype: IgG2b, κ, Reference: 563126), was employed at a dilution of 1/30. Control conditions comprised Isotype + FMO and Providers + NHP reagent. CD103, labeled with BV650 and sourced from BD Biosciences (Clone: Ber-ACT8, Isotype: IgG1, κ, Reference: 743653), was used at a dilution of 1/20. Control conditions included Isotype + FMO and Providers + NHP reagent. KI-67, labeled with AF®488 and obtained from BD Biosciences (Clone: B56, Isotype: IgG1, κ, Reference: 561165), was utilized at a dilution of 1/20. Control conditions comprised Isotype + FMO and Providers + NHP reagent. GZMB, labeled with V450 and provided by BD Biosciences (Clone: GB11, Isotype: IgG1, Reference: 561151), was used at a dilution of 1/15. Control conditions included Isotype + FMO and Providers + NHP reagent. IFN-γ, labeled with BV510 and sourced from BD Biosciences (Clone: B27, Isotype: IgG1, κ, Reference: 563287), was employed at a dilution of 1/20. Control conditions comprised Isotype + FMO and Providers + NHP reagent. IL-17, labeled with BV786 and obtained from BioLegend (Clone: BL168, Isotype: IgG1, Reference: 512337), was utilized at a dilution of 1/20. Control conditions included Isotype + FMO and Providers + NHP reagent. CD4, labeled with PE-Cy™7 and sourced from BD Biosciences (Clone: L200, Isotype: IgG1, κ, Reference: 560644), was used at a dilution of 1/20. Control conditions comprised Isotype + FMO and Providers + NHP reagent. CD8, labeled with VioBlue® and obtained from Miltenyi Biotec (Clone: Bw135/80, Isotype: IgG2aκ, Reference: 130-113-162), was employed at a dilution of 1/30. Control conditions included Isotype + FMO and Providers + NHP reagent. CD7, labeled with BV650 and provided by BD Biosciences (Clone: M-T701, Isotype: IgG1, κ, Reference: 740565), was used at a dilution of 1/15. Control conditions comprised Isotype + FMO and Providers + NHP reagent. HLA-DR, labeled with AF®700 and sourced from BD Biosciences (Clone: L243 (G46-6), Isotype: IgG2a, κ, Reference: 560743), was utilized at a dilution of 1/20. Control conditions included Isotype + FMO and Providers + NHP reagent. CD20, labeled with BV711 and obtained from BD Biosciences (Clone: 2H7, Isotype: IgG2b, κ, Reference: 563126), was employed at a dilution of 1/20. Control conditions comprised Isotype + FMO and Providers + NHP reagent. CD45RA, labeled with BV786 and sourced from BD Biosciences (Clone: 5H9, Isotype: IgG1, κ, Reference: 741010), was used at a dilution of 1/20. Control conditions included Isotype + FMO and Providers + NHP reagent. CD34, labeled with BUV737 and provided by BD Biosciences (Clone: 563, Isotype: IgG1, κ, Reference: 741868), was utilized at a dilution of 1/15. Control conditions comprised Isotype + FMO and Providers + NHP reagent. EOMES, labeled with FITC and sourced from Invitrogen (Clone: WD1928, Isotype: IgG1, κ, Reference: 11-4877-42), was used at a dilution of 1/30. Control conditions included Isotype + FMO and Providers + NHP reagent. T-bet, labeled with PerCP/Cy5.5 and obtained from SONY (Clone: 4B10, Isotype: IgG1, κ, Reference: RT3824030), was employed at a dilution of 1/10. Control conditions comprised Isotype + FMO and Providers + NHP reagent. IL-21, labeled with AF®647 and provided by BD Biosciences (Clone: 3A3-N2.1, Isotype: IgG1, Reference: 562043), was used at a dilution of 1/40. Control conditions included Isotype + FMO and Providers + NHP reagent. IFN-γ, labeled with APC/Cy7 and sourced from SONY (Clone: B27, Isotype: IgG1, κ, Reference: RT3132620), was employed at a dilution of 1/15. Control conditions comprised Isotype + FMO and Providers + NHP reagent. CD20, labeled with A700 and obtained from BD Biosciences (Clone: 2H7, Isotype: IgG2b, κ, Reference: 560631), was used at a dilution of 1/30. Control conditions included Isotype + FMO and Providers + NHP reagent. CD34, labeled with BUV737 and provided by BD Biosciences (Clone: 563, Isotype: IgG1, κ, Reference: 741868), was utilized at a dilution of 1/20. Control conditions comprised Isotype + FMO and Providers + NHP reagent. CD45RA, labeled with BV786 and sourced from BD Biosciences (Clone: 5H9, Isotype: IgG1, κ, Reference: 741010), was used at a dilution of 1/20. Control conditions included Isotype + FMO and Providers + NHP reagent. CCR7, labeled with APC/Cy7 and sourced from BioLegend (Clone: G043H7, Isotype: IgG2a, κ, Reference: 353211), was utilized at a dilution of 1/15. Control conditions included Isotype + FMO and Providers + NHP reagent. CXCR3 (CD183), labeled with PE-Cy™7 and obtained from BD Biosciences (Clone: 1C6, Isotype: IgG1, κ, Reference: 560831), was employed at a dilution of 1/30. Control conditions comprised Isotype + FMO and Providers + NHP reagent. CD44, labeled with BUV661 and provided by BD Biosciences (Clone: G44-26, Isotype: IgG2b, κ, Reference: 741615), was used at a dilution of 1/20. Control conditions included Isotype + FMO and Providers + NHP reagent. CD127, labeled with PE-Cy5 and sourced from Invitrogen (Clone: eBioRDR5, Isotype: IgG1 κ, Reference: 15-1278-42), was employed at a dilution of 1/20. Control conditions comprised Isotype + FMO and Providers + NHP reagent. GATA-3, labeled with BV421/450 and obtained from BD Biosciences (Clone: L50-823, Isotype: IgG1, κ, Reference: 563349), was utilized at a dilution of 1/20. Control conditions included Isotype + FMO and Providers + NHP reagent.

T-bet, labeled with PerCP/Cy5.5 and sourced from SONY (Clone: 4B10, Isotype: IgG1, κ, Reference: RT3824030), was used at a dilution of 1/20. Control conditions comprised Isotype + FMO and Providers + NHP reagent. CD14, labeled with alexa 488, FITC, and BB515, was obtained from Miltenyi Biotec (Clone: TÜK4, Isotype: IgG2aκ, Reference: 130-113-146). It was used at a dilution of 1/10, and control conditions included Isotype + FMO and Providers + NHP reagent. MHC-E, labeled with PE and provided by NOVUS (Clone: 3D12MHLA-E, Isotype: IgG1, κ, Reference: NBP2-00277), was used at a dilution of 1/30. Control conditions comprised Isotype + FMO and Providers + NHP reagent. CD206, labeled with ECD and sourced from BD Biosciences (Clone: 19.2, Isotype: IgG1), was employed in our analysis. Control conditions included Isotype + FMO and Providers + NHP reagent. IL-10, labeled with PC7 and obtained from BioLegend (Clone: JES3-9D7, Isotype: IgG1, κ, Reference: 501419), was utilized in our experiments. Control conditions comprised Isotype + FMO and Providers + NHP reagent. CD64, labeled with APC and provided by BD Biosciences (Clone: 10,1, Isotype: IgG2b, κ, Reference: 561189), was used in our analysis. Control conditions included Isotype + FMO and Providers + NHP reagent. CD11C, labeled with BV605 and sourced from BD Biosciences (Clone: S-HCL-3, Isotype: IgG2b, κ, Reference: 744436), was employed in our experiments. Control conditions comprised Isotype + FMO and Providers + NHP reagent. NKG2c, labeled with Vio® Bright R720 and obtained from Miltenyi Biotec (Clone: REA205, Isotype: IgG2b, κ, Reference: 130-130-663), was utilized in our analysis. Control conditions included Isotype + FMO and Providers + NHP reagent. CD163, labeled with BV421/450 and provided by BioLegend (Clone: GHI/61, Isotype: IgG1), was used in our experiments. Control conditions comprised Isotype + FMO and Providers + NHP reagent.

Validation

the antibodies employed for flow cytometry analysis underwent a thorough validation process to ensure their reliability and accuracy in detecting specific immune cell populations and markers. The validation process included several critical steps:

Antibody Selection: The initial step in the validation process involved the careful selection of antibodies. Antibodies were chosen

based on their known specificity for the target antigens of interest. Extensive literature review and consultation with experts in the field were conducted to confirm their suitability for our study.

Manufacturer's Specifications: We reviewed the manufacturer's specifications and technical data sheets for each antibody. This information included details such as clone names, isotypes, recommended dilutions, and references. Understanding these specifications was essential for proper antibody usage.

Positive and Negative Controls: To validate antibody performance, we used positive and negative controls. Positive controls included cells or tissues known to express the target antigen, while negative controls were cells or tissues lacking the antigen. These controls helped confirm the specificity of the antibodies.

Fluorescence Minus One (FMO) Controls: FMO controls were utilized to assess background fluorescence and determine the true signal from the target antigen. FMO controls included all fluorochromes except the one specific to the tested antibody, allowing us to differentiate specific staining from non-specific background fluorescence.

Non-Human Primate (NHP) Reagent Controls: Since our study involved non-human primates, NHP reagent controls were used to ensure the compatibility and specificity of the antibodies in the primate model. These controls helped verify that the antibodies effectively bound to the target antigens in our experimental system.

Dilution Optimization: Antibody dilutions were optimized to achieve the best signal-to-noise ratio. Serial dilutions of each antibody were tested to determine the optimal concentration for accurate and robust staining.

# Eukaryotic cell lines

Policy information about cell lines

| Cell line source(s) | K562 cell line was provided by the ATCC ; The K-562 cell line was purchased from ATCC and transduced with a lentivirus expression vector for human HLA-E was purchases from Applied Biological Materials Inc. |
| --- | --- |
| Authentication | None of the cell lines were authenticated. |
| Mycoplasma contamination | We confirm that the cell line was tested negative for mycoplasma |
| Commonly misidentified lines (See ICLAC register) | ATCC® Number: CCL-243™ ; no commonly misidentified cell lines were used in the study. |

# Animals and other organisms

Policy information about studies involving animals; ARRIVE guidelines recommended for reporting animal research

| Laboratory animals | Cynomolgus macaques (M. fascicularis) aged 37–60 months were sourced from Mauritian AAALAC-certified breeding centers. The study was conducted at IDMIT infrastructure facilities with appropriate containment levels in accordance with regulations. At the time of infection, all animals were seronegative for specific viruses. Following infection, the macaques were housed individually in biosafety facilities. Both male and female monkeys were included in the study, and sample collection was performed randomly |
| --- | --- |
| Wild animals | Study did not involve wild animals. |
| Field-collected samples | Study did not involve field-collected samples |
| Ethics oversight | The study protocols received approval from the institutional ethical committee, 'Comité d'Ethique en Expérimentation Animale du Commissariat à l'Energie Atomique et aux Energies Alternatives' (CEtEA 44), under statement numbers A20_011, 20_066, and 21_069. Authorization for the study was granted by the 'Research, Innovation, and Education Ministry' under registration numbers APAFIS#24434–2020030216532863v1, APAFIS#29191-2021011811505374 v1, and APAFIS#33414-2021101115102064 v1. |

Note that full information on the approval of the study protocol must also be provided in the manuscript.

# Human research participants

Policy information about studies involving human research participants

| Population characteristics | Blood samples was collected based on voluntary. To be allowed to give blood sample, Volunteers should had between 18 and 70 years old, and weigh more than 50 kg. Exact age, sexe and ethnic origin of the participantes is not known. Fifty milliliter blood samples were obtained from twelve donors. |
| --- | --- |
| Recruitment | Blood samples was collected based on voluntary. All blood samples were collected before 2019 and PBMC were isolated using Ficoll gradient and immediately frozen. |
| Ethics oversight | Blood samples from healthy donors were obtained from the French blood bank (Etablissement Français du Sang) as part of an agreement with the Institut Pasteur (C CPSL UNT, number 15/EFS/023). The study was approved by the Ethics Review Committee (Comité de protection des personnes) of Île-de-France VII. |

Note that full information on the approval of the study protocol must also be provided in the manuscript.

# Flow Cytometry

## Plots

Confirm that:

☒ The axis labels state the marker and fluorochrome used (e.g. CD4-FITC).

☒ The axis scales are clearly visible. Include numbers along axes only for bottom left plot of group (a 'group' is an analysis of identical markers).

☒ All plots are contour plots with outliers or pseudocolor plots.

☒ A numerical value for number of cells or percentage (with statistics) is provided.

## Methodology

Sample preparation

Whole venous blood was collected in ethylenediaminetetraacetic acid (EDTA) tubes. Peripheral blood mononuclear cells (PBMCs) were isolated by Ficoll density-gradient centrifugation. Broncho alveolar lavage fluid have been collected when bllod puncture was done. The cell suspension was subsequently filtered through 100- and 40-µm cell strainers, and cells were washed with cold phosphate-buffered saline (PBS). Cells were either immediately stained for flow cytometry or cryopreserved in 90% foetal bovine serum (FBS) and 10% dimethyl sulfphoxide (DMSO) and stored in liquid nitrogen before use. Intracellular staining was performed using BD Cytofix/Cytoperm™.

Instrument

Flow cytometry acquisitions were done on a LSRFortessa (BD Biosciences).

Software

The data were analysed using Diva (software version 6.1.3, build 2009 05 13 13 29). The data were further analyzed using FlowJo (10.4.2 software, LLC, Ashland, OR, USA). Multiparametric analyses were performed using SPICE (version 5.1). t-SNE was performed with the cytobank (Cytobank, Inc.), using 2,000 iterations and a perplexity of 60.

Cell population abundance

Online methods, paragraph: Polychromatic flow cytometry ; Cell sorting of NK cell sub-populations.

Gating strategy

all gating strategies are provided in extended figures

☒ Tick this box to confirm that a figure exemplifying the gating strategy is provided in the Supplementary Information.

