## [Peer Review File · Nature Immunology]

Peer Review Information

Journal: Nature Immunology

Manuscript Title: SARS-CoV-2 viral persistence in lung alveolar macrophages is controlled by IFN- γ and NK cells

Corresponding author name(s): Dr. Nicolas Huot

Reviewer Comments & Decisions:

Decision Letter, initial version:
--

6th Jun 2022

Dear Dr. Huot,

We have now finished reviewing your manuscript entitled "Adaptive MHC-E restricted tissue-resident NK cells are associated with persistent low antigen load in alveolar macrophages after SARS-CoV-2 infection", reference number NI-A33951-T. While the work is of potential interest, the reviewers have raised substantial concerns. As such, we cannot accept the current manuscript for publication.

Should you find yourself able to thoroughly address the referees' concerns, please let me know. If the novelty of the paper has not been compromised in the interim, it is possible that you could resubmit a new paper, although we hope you understand that until we have read the revised manuscript in its entirety we cannot promise that it will be sent back for peer review.

If you decide to resubmit, please include a "Response to referees" detailing, point-by-point, how you addressed each referee comment. This response will be sent back to the referees along with the revised manuscript.

Please include a revised version of any required reporting checklist. It will be available to referees (and, potentially, statisticians) to aid in their evaluation if the manuscript goes back for peer review. A revised checklist is essential for re-review of the paper. The Reporting summary can be found here: <https://www.nature.com/documents/nr-reporting-summary.pdf>

In addition, please pay close attention to our [href="https://www.nature.com/nature-research/editorial-policies/image-integrity">Digital Image Integrity Guidelines. and to the following points below:](https://www.nature.com/nature-research/editorial-policies/image-integrity)

-- that unprocessed scans are clearly labelled and match the gels and western blots presented in figures.

- that control panels for gels and western blots are appropriately described as loading on sample processing controls
- all images in the paper are checked for duplication of panels and for splicing of gel lanes.

Nature Immunology is committed to improving transparency in authorship. As part of our efforts in this direction, we are now requesting that all authors identified as 'corresponding author' on published papers create and link their Open Researcher and Contributor Identifier (ORCID) with their account on the Manuscript Tracking System (MTS), prior to acceptance. This applies to primary research papers only. ORCID helps the scientific community achieve unambiguous attribution of all scholarly contributions. You can create and link your ORCID from the home page of the MTS by clicking on 'Modify my Springer Nature account'. For more information please visit www.springernature.com/orcid.

Please do not hesitate to contact me if you have any questions. Thank you for the opportunity to review your work.

Sincerely,

Ioana Visan, Ph.D.
Senior Editor
Nature Immunology

Tel: 212-726-9207
Fax: 212-696-9752
www.nature.com/ni

Reviewers' comments:

Reviewer #1 (Remarks to the Author):

The authors characterize SARS-CoV-2 infection in cynomolgus macaques with a specific interest in the persistence of viral antigens over prolonged periods of time. Interestingly, SARS-CoV-2 protein expression was detected in alveolar macrophages beyond 18 months after the infection. Multi (12)-parametric flow cytometry analysis of BALF macrophages revealed two groups displaying an activated (group 1) and a more regulatory/M2 (group 2) marker constellation, respectively and group 2 was associated with a lower virus load. Within group 2, IL-10 and HLA-E expression were closely linked. As HLA-E is a ligand for NK cell receptors (i.e., NKG2A and NKG2C) and NK cells were increased in numbers in group 2, the authors characterized NK cell responses. In group 2 animals, higher NK cell numbers in BALF were associated with a higher fraction of NK cells expressing CXCR3, CCR7, CD69, CD107a and Ki67. In contrast, less NK cells expressed IFN γ in group 2 the frequency of which correlated positively with spike+ macrophages. Finally, the authors identified spike-derived peptides that via HLA-E inhibited NK cell degranulation of human NKG2A+ NK cells and of convalescent group 1

monkey NK cells. However, BALF NK cells from group 2 were activated by HLA-E:peptide (and spike peptides from other Coronaviruses) in vitro suggesting adaptive re-programming of NK cells.

This is an interesting manuscript that is very well written. Overall, the data is of good quality. The observation of long-term virus persistence in macrophages is interesting and the model, that the degree of virus persistence is dependent on NK cell states and a switch in the usage of NKG2 receptors is an interesting one. Having said that there are also significant issues with the data presented that are summarized below.

1. The data show persistence of SARS-CoV-2 in macaques beyond 18 months after the initial infection. While this is a thought-provoking finding, its relevance for human infections remains unclear. Are human (BALF) macrophages also persistently infected with SARS-CoV-2? Such data should be added to the manuscript.
2. As all animals resolved the infection the relevance of (i) spike+ macrophages in general and of (ii) rather small differences in the frequency of spike+ macrophages is questionable. Are there any relevant differences in disease burden or sequelae after the infection in monkeys that support relevant consequences? Is there persistence of infectious virus or not?
3. The identification of SARS-CoV-2-derived HLA-E-restricted peptides that inhibit NK cells is surprising as other groups demonstrated that peptides identified based on the SARS-CoV-2 ORFome are not inhibiting NK cells via NKG2A (PMID: 35235832) and that MHC-I blockade did not enhance NK cell-mediated control of SARS-CoV-2 replication (PMID: 34695836). How do the authors explain these discrepancies?
4. Authors make the curious observation that NK cells from group 1 are inhibited by HLA-E:spike peptides whereas NK cells from group 2 are activated. This is mechanistically not explored. Is there a change in the representation of NKG2A (inhibitory) and NKG2C (activating) in NK cells from group 1 vs. group 2? This is a critical limitation for the interpretation of the data and this should be explored by additional experiments. Specific Abs for NKG2A or NKG2C could be used or single cell transcriptional profiling (by PCR) of NK cells.
5. The authors show the impact of HLA-E-restricted peptides on NK cell degranulation (measured as surface CD107 expression). This is not a direct proof for target cell lysis because degranulation can occur without killing (e.g., if cytotoxic machinery is lacking). To directly link NK cells to restricting the frequency of spike+ macrophages, viral load or spike protein expression in infected cells after co-culture with group 1 or group 2 NK cells or assays for target cell killing (chromium release assays) should be used.
6. The axis labels of "specific lysis" (Figure 5) are misleading as CD107a expression is measured and not target cell lysis. What exactly do the values on the y-axis indicate?

Minor points

1. Some of the animals have been treated with hydroxychloroquine. Is there any correlation of this treatment with the phenotypes for macrophages or NK cells described by the authors?

2. The animals have been sampled for BALF between 6 and 18 months after SARS-CoV-2. How many times have the individual monkeys been sampled? The specific time points after the infections should be indicated in Table 1. Is there any correlation of the time point after infection when the cells have been analyzed vs. viral load in macrophages or phenotype observed for macrophages / NK cells?
3. There is a lot of redundancy in the data shown in all main figures. Cluster analysis that recapitulates heterogeneity that has already been shown for individual markers in preceding panels should be transferred into the supplement along with part of the negative data.
4. The manual gate for SARS-CoV-2+ macrophages is not shown.

Reviewer #2 (Remarks to the Author):

The study by Huot et al. explores the mechanisms that enable the long-term persistence of SARS-CoV2 virus in infected non-human primates. They show that a subset of convalescent macaques harbor infected alveolar macrophages (AM) for prolonged periods of time. They also claim that infected AM in these animals persist due to HLA-E mediated inhibition of NK cell function.

This study is very difficult to read and the figures are poorly presented making it very difficult to fully assess data quality. Many claims are overstated including the claim that infected macrophages have a regulatory phenotype. The phenotype of NK cells is very limited likely due to the lack of NHP reagents.

The authors show that NK cells isolated from the BAL of viral persistent animals do not express CD107a when cocultured with lines pulsed with specific SARS CoV2 encoding leader sequences of SPIKE proteins, which the authors claim are preferentially presented in MHC- E grooves, whereas NK cells isolated from animals that do not harbor persistent infected AM retain CD107a and claim that persistence of infected AM might be due to inhibition of NKG2a+ NK effector function through engagement of NKG2a inhibitory function.

However the rationale for targeting the leader sequences of SPIKE proteins across the coronaviruses does not acknowledge how antigen processing works and ignores the rest of the genome that could potentially derive peptides with suitable biochemistry for MHC-E binding. It seems the authors chose the leader sequences of Spike solely because of the HLA-ABC derived peptide (VL9, VMAPRTLLL) that binds to HLA-E. It ignores what we know about UL-40-derived HLA-E-binding peptides as well as other emerging viruses, including SARS-CoV-2 (Hammer et al, Cell Reports, 2022). In the Hammer study, the authors identified SARS CoV2 encoded HLA-E restricted peptides that do not bind NKG2a inhibitor receptor and that enable NK cell activation, highlighting the diversity of viruses encoded peptides and their ability to train NK cell function

Overall while the observation that a subset of NHP can harbor infected AM for very prolonged periods of time the mechanisms that enable the persistence of infected AM is unclear. In addition it would have been interesting to explore the clinical consequence of infected AM.

Reviewer #3 (Remarks to the Author):

Huot et al use a non-human primate model (cynomolgus macaques) to study SARS-CoV-2 infection,

with particular focus on macrophage and NK cell responses in the lung. Using this model, the authors describe persistence of SARS-CoV-2 antigen in alveolar macrophages up to at least 18 months after infection, and an association between lower levels of persisting SARS-CoV-2 antigen and spike-specific HLA-E restricted NK cells in the lung. Overall, this is an interesting observation and the data are well presented. Given the observation that SARS-CoV-2 antigen can persist in some humans, a better understanding of the mechanisms and consequences of viral antigen persistence is important. Furthermore, the interactions between HLA-E and NKG2-receptors in SARS-CoV-2, and their relevance for disease outcome, are not well understood; with a recent study suggesting that SARS-CoV-2 encodes for an HLA-E stabilizing peptide that can abrogate inhibition of NKG2A-expressing human NK cells (Hammer et al, Cell Rep 2022). Despite the importance of the topic, there are a number of major concerns that reduce the overall significance of the presented findings.

Major concerns:

- A central observation of the study by Huot et al is that SARS-CoV-2 RNA and Spike protein were detected in BAL of all infected macaques 6-18 months post infection. These data are of interest, as they suggest that SARS-CoV-2 can persist in non-human primates for extended period of time. However, such long-term persistence has not been described for most humans, raising concerns on how representative the macaque model used here is for human SARS-CoV-2 infection, and the authors should address this important limitation more clearly.
- Huot et al show that macrophages in BAL differed between healthy and convalescent macaques, and that two groups of convalescent macaques can be distinguished based on macrophage phenotypes. Convalescent macaques with a more regulatory macrophage phenotype (group 2), including elevated MHC-E and IL-10 expression, also exhibited lower levels of Spike antigen and lower levels of SARS-CoV-2 VL in BAL. NK cells from macaques included in group 2 were enriched in BAL, and these NK cells expressed higher levels of tissue homing receptors. Furthermore, NK cells from animals classified as group 2 displayed higher differentiation and were associated negatively with SARS-CoV-2 antigen load. The authors show many correlative data demonstrating differences in macrophage and NK cell phenotypes between macaques with higher and lower persistence of SARS-CoV-2 antigen. The analysis is complicated by many sub-analyses, and the actual differences between the groups, in particular for NK cells, is often marginal and not statistically significant. Furthermore, it appears to the reviewer that the statistical analysis was not adjusted for the (many) multiple comparisons performed. Overall, the analysis indicate differences in macrophage and NK cell populations between macaques with higher and lower antigen persistence in the BAL, but it remains unclear whether such differences are solely the consequences of differences in antigen load, or whether immunological differences contribute to differences in viral load. Finally, no information on T cell and B cell populations, as well as antibody levels (all of which have been associated with viral control in humans infected with SARS-CoV-2) are provided.
- In the last part of the manuscript, the authors identify HLA-E binding peptides within CoV, based on their binding motif, and assess their impact on NK cell function after loading MHC-E expressing cells with the peptides. The relevance of these studies is limited by a number of caveats. The activating NKG2C and the inhibitory NKG2A receptor can bind to HLA-E presenting peptides, but these receptors cannot be distinguished in macaques, limiting the conclusions regarding NKG2A/C interactions with MHC-E that can be drawn from these experiments. Furthermore, the tested peptides varied in their ability for stabilizing MHC-E, and NK cell killing of target cells is not adjusted for these differences. As peptide-prediction approaches were used, it remains unknown whether any of the investigated peptides is actually presented on SARS-CoV-2-infected MHC-E+ macrophages. Therefore, it is unclear

whether in vitro observations on NK cell lysis using high concentrations of peptides might actually reflect in vivo situations. It would be important to determine whether SARS-CoV-2 antigens are actually presented by MHC-E on Spike+ macrophages, for examples using MS. Finally, it remains unclear why NK cells derived from group1 and group2 animals differ in their response to peptide-pulsed MHC-E expressing target cells. No potential mechanism is provided.

Additional points:

- The authors state that early imprinting and training of macrophages and NK cells might explain their observations – however, SARS-CoV-2 antigen load differs between the two groups of animals at the time of analysis, suggesting that differences in antigen load might drive the observed differences in macrophage and NK cell phenotypes. This is further suggested by the observed differences between Spike-positand Spike-neg macrophages.
- No mechanisms is provided on how “ NK cells from about half of the animals were capable to adapt and were not any more inhibited by Spike LS peptides” (page12). If this were indeed the case, it would be important to understand the underlying mechanism to increase the significance of the described observations.
- A recently published study by Hammer et al (Cell Rep 2022) described that SARS-CoV-2 encodes for an HLA-E-stabilizing peptide that abrogates inhibition of NKG2A+ NK cells. How can these observations in humans be reconciled with the data described here using a macaque model?

Decision Letter, first revision:

1st Jul 2023

Dear Dr. Huot,

Thank you for your response to the referees' comments on your manuscript "Regulation by NK cells and IFN- γ of SARS-CoV-2 viral persistence in alveolar macrophages of the lung". Although we are interested in the possibility of publishing your study in Nature Immunology, the issues raised by the referees need to be addressed.

Please revise along the lines specified in your letter. Please clarify how many samples had detectable virus in the lung macrophages ex vivo and how many after culture and how is this linked to NK cells activation/ability to control the virus in the respective monkeys. These separate comparisons, as requested by referee #2, should be added in the manuscript. At resubmission, please include a "Response to referees" detailing, point-by-point, how you addressed each referee comment. If no action was taken to address a point, you must provide a compelling argument. This response will be sent back to the referees along with the revised manuscript.

Please include a revised version of any required reporting checklist. It will be available to referees to aid in their evaluation. The Reporting Summary can be found here:
<https://www.nature.com/documents/nr-reporting-summary.pdf>

When submitting the revised version of your manuscript, please pay close attention to our [href="https://www.nature.com/nature-portfolio/editorial-policies/image-integrity">Digital Image Integrity Guidelines.](https://www.nature.com/nature-portfolio/editorial-policies/image-integrity) and to the following points below:

[REDACTED]

We hope to receive your revised manuscript within two weeks. If you cannot send it within this time, please let us know. We will be happy to consider your revision so long as nothing similar has been accepted for publication at Nature Immunology or published elsewhere.

Nature Immunology is committed to improving transparency in authorship. As part of our efforts in this direction, we are now requesting that all authors identified as 'corresponding author' on published papers create and link their Open Researcher and Contributor Identifier (ORCID) with their account on the Manuscript Tracking System (MTS), prior to acceptance. ORCID helps the scientific community achieve unambiguous attribution of all scholarly contributions. You can create and link your ORCID from the home page of the MTS by clicking on 'Modify my Springer Nature account'. For more information please visit www.springernature.com/orcid.

Sincerely,

Ioana Visan, Ph.D.
Senior Editor
Nature Immunology

Tel: 212-726-9207
Fax: 212-696-9752
www.nature.com/ni

Reviewers' Comments:

Reviewer #1:

Remarks to the Author:

This is a revised version of this manuscript that has addressed most of the technical concerns raised in the previous round of review. The overall observation remains interesting but the revised manuscript still does not give much mechanistic insights into the molecular networks allowing for virus persistence and how this possibly relates to long term sequelae of COVID-19.

Reviewer #2:

Remarks to the Author:

The authors provide strong evidence that in SARSCoV2 infected macaques, CoV2 virus can persist for very prolonged periods (up to 18 months) in alveolar macrophages and that viral persistence can contribute to cell-cell viral spreading. However the authors fail to identify clinical consequences of the viral persistence- the macaques did not develop any overt clinical symptoms and symptoms observed in patients with long COVID were not collected or reported. The authors claim that viral persistence is associated with increased cytokine release in the blood circulation, however with the exception of IL-23, the levels of IL-6 and IL-18, IL-15 and IP10 are very low and unlikely pathogenic. It is unfortunate that the authors did not spend more time to establish whether viral persistence could contribute to clinical symptoms observed in humans with long COVID as this would have increased the relevance and impact of the study

The authors also tried to link Cov2 persistence to defective NK cell function triggered by MHC-E dependent inhibition of NKG2a+ NK cells but here again the data lack strength as changes in phenotypes and function are modest- a better approach would have been to isolate NK cells and CD8+ T cells from convalescent macaques that have or do not exhibit viral persistence and compare their ability to eliminate virally infected cells ex vivo.

Reviewer #3:

Remarks to the Author:

The authors addressed all the concerns I raised, both by performing additional experiments/analyses, and by discussing remaining limitations. Congratulations to a very interesting study.

Author Rebuttal, first revision:

Point per point response to the reviewer's comments

(Comments by the reviewers are in blue, and answers in black)

Reviewer #1

This is a revised version of this manuscript that has addressed most of the technical concerns raised in the previous round of review. The overall observation remains interesting, but the revised manuscript still does not give much mechanistic insights into the molecular networks allowing for virus persistence and how this possibly relates to long term sequelae of COVID-19.

In the manuscript we provide several novel mechanistic insights into viral persistence and NK cell function during SARS-CoV-2 infection. We provide mechanistic insights into the virus spreading between macrophages (long-awaited in the literature), identified a novel MHC-E dependent regulation of NK cell activity, unraveled a rheostat role of IFN- γ in the control of viral replication in tissue macrophages, and a mechanism of how this IFN- γ production could be reduced in NK cells after SARS-CoV-2 infection.

In the newly revised manuscript, we added additional analyses on the existing data about the mechanistic links with viral persistence and additional explanations in the text (see answer to reviewer 2 for details, and figures 2c,2f, fig.3b, d, fig.4 and Fig 5b, k and associated supplementary figures).

The reviewer is correct in stating that our observations are difficult to link to symptoms described for long-COVID patients largely due to the fact that most of the symptoms used to define long-COVID status are not experimentally measurable or quantifiable but are descriptive from individual patients following SARS-CoV-2 infection (i.e extreme tiredness (fatigue), feeling short of breath, loss of smell, muscle aches, problems with memory and concentration ("brain fog"), chest pain or tightness, difficulty sleeping (insomnia), heart palpitations, depression and anxiety, etc. Moreover, the number of symptoms, their intensity, and how they are scored vary widely by country, the composition of the cohort, the individual etc. There is no biological, immunological or virological biomarker for long-COVID thus it would be difficult if not impossible to define it in an experimental animal model. We understand the reviewer's interest here but feel this goes beyond the scope of a manuscript like ours that is not clinical in nature but focused on specific biological mechanisms of viral persistence. Specifically, the study characterizes the mechanism of long-term viral persistence in tissues, the impact of SARS-CoV-2 infection on NK cells and the role of NK cells on viral load and long-term viral persistence in tissues. There is no doubt any more in the literature, that SARS-CoV-2 can persist in humans, which is a major issue per se.

Reviewer #2

The authors provide strong evidence that in SARS-CoV-2 infected macaques, the virus can persist for very prolonged periods (up to 18 months) in alveolar macrophages and that viral persistence can contribute to cell-cell viral spreading. However, the authors fail to identify clinical consequences of the viral persistence- the macaques did not develop any overt clinical symptoms and symptoms observed in patients with long COVID were not collected or reported. The authors claim that viral persistence is associated with increased cytokine release in the blood circulation, however with the exception of IL-23, the levels of IL-6 and IL-18, IL-15 and IP10 are very low and unlikely pathogenic. It is unfortunate that the authors did not spend more time to establish whether viral persistence could contribute to clinical symptoms observed in humans with long COVID as this would have increased the relevance and impact of the study.

As mentioned above, our observations are difficult to link to symptoms described for long-COVID patients mainly due to the fact that most of the symptoms used to define long-COVID status are not experimentally measurable or quantifiable but are descriptive from individual patients following SARS-CoV-2 infection. Moreover, the number of symptoms, their intensity, and how they are scored vary widely by country, the composition of the cohort, etc. There is no biological, immunological or virological biomarker for long-COVID and thus it would be difficult if not impossible to define it in an experimental animal model. We understand the Reviewer's interest here but feel this goes beyond the scope of a manuscript like ours that is not clinical in nature but focused on biological and immunological mechanisms of viral persistence.

We agree with the reviewer on the comment about the cytokines. However, we challenge the notion of "unlikely pathogenic" used by the reviewer, regarding the levels of cytokines. Indeed, in the HIV field for example, a residual low level but chronically persisting inflammation is associated with a higher risk of long-term noncommunicable diseases. Moreover, even if the levels of IL-6 and IL-18, IL-15 and IP10 were low, this does not mean that the low-level measures in plasma reflect necessarily the amount of corresponding cytokine in tissues.

Indeed, elevated levels of various pro-inflammatory cytokines have been observed in some individuals with long-COVID. However, also in Long Covid patients, there are contradictory data on cytokine profiles and no clear evidence that cytokine levels are increased to pathogenic levels in long-COVID. Moreover, one study on Long Covid patients even reported decreased cytokine levels. Of note, we observed that IL-6 and IP-10 levels positively correlated with the percentage of macrophages harboring virus (page 5 and Supplementary Fig. 1). This is a key finding. In addition, higher IL-23 levels have been associated in the literature with increased disease severity and prolonged recovery.

Thus, our results are not discordant with other studies in humans and on long-COVID patients. Of note, the scope of the study was not to understand long-COVID, but to understand mechanisms of long-term viral persistence (unraveling cell-to-cell spreading among macrophages in lung-associated tissues) and the role of NK cells and IFN- γ on regulating long-term viral persistence in tissues. There is no doubt any more in the literature, that SARS-CoV-2 can persist in humans, which is a major issue per se.

In order to facilitate the interpretation of the plasmatic cytokine levels, in particular it's link with viral persistence, and to answer to the reviewer's comment, and we have edited some figures of the manuscript by clustering the monkeys according to the level of viral RNA in BALF cells as described in the revised manuscript (Supplementary Figure 5, pages 6) and below. One group (N=8 monkeys) was composed of animals with high levels of persisting viral load, another group (N=9 monkeys) with low viral load close to threshold, and a third group (N=8) animals with levels below the threshold (Supplementary Figure 5 a). We also added a quantification in plasma of all animals

of a soluble marker of monocyte/macrophage activation (sCD14). By doing this, we noticed that the level of several inflammatory markers (ie IP-10, sCD14) was higher in the group of monkeys harboring high viral RNA when compared to the other groups. These new analyses are included in the manuscript in the Supplementary Figure 5.

The authors also tried to link Cov2 persistence to defective NK cell function triggered by MHC-E dependent inhibition of NKG2a+ NK cells but here again the data lack strength as changes in phenotypes and function are modest- a better approach would have been to isolate NK cells and CD8+ T cells from convalescent macaques that have or do not exhibit viral persistence and compare their ability to eliminate virally infected cells *ex vivo*.

We are not sure what the reviewer means by “modest” changes. Indeed, the correlations for example between the capacity of NK cell inhibition and viral persistence are strong (i.e. Figure 5i, $p > 0.0001$ and $p = 0.0014$ for levels of BALF viral RNA and BALF Spike⁺macrophages, respectively).

To address the reviewers' questions, we have made several revisions to the manuscript. We now compare the animals also based on the viral load measured in BAL. As suggested by the reviewer, we compared convalescent macaques with different degrees of viral persistence (no, intermediate, high). To achieve this new classification, we divided the monkeys into three groups (Supplementary Figure 5) based the quantity of viral RNA in BALF cells (Figure 2a). The first group comprises eight animals characterized by a high level of persisting viral load. The second group consists of nine animals with a detectable, but very low viral load, , and the third group includes eight animals with undetectable viral RNA in BALF cells.

Based on this revised animal classification, we provide several new comparisons in the revised manuscript. As mentioned above (response to reviewer #1), we show that systemic markers of inflammation were higher in animals with a high viral load compared to the other groups of animals (see Supplementary Figure 5).

Among the other additional analyses, we edited Figure 2 f (previously Figure 2g) to analyze the level of Spike production in macrophages during in vitro culture over time depending on the initial viral RNA levels (page 7)

We furthermore revised the analyses of the lymphocytes, including NK cells and T cells (Figure 3 b, d, e, f of the revised manuscript).

We demonstrate that animals with a high viral load exhibit the lowest capacity of NK cells and NKG2r+CD8+ T cells to produce IFN- γ , in contrast to animals with a low or undetectable viral load, which have high levels of IFN- γ (Figure 3d, e, f and Figure 4 g. This new observation suggests that NKG2r+CD8+ T cells and NK cells, mainly the NKG2A^{low} NK cells (Figure 5 c and d), play an important role in viral persistence. Indeed, we show that IFN-g inhibits viral production in the BALF macrophages. However, through this mechanism, the infected cell is not eliminated and the virus can persist and eventually be re-activated. This link is emphasized in the manuscript by the strong negative correlations of both IFN- γ + NK cells and NKG2r+CD8+ T cells with the *ex vivo* levels of Spike+ macrophages (Figure 3g of the revised manuscript).

To further support our observation regarding the role of IFN- γ in viral replication in macrophages, we have moved the results regarding the impact of IFN- γ on viral replication and MHC-E expression from Figure 5 of the previous version of the manuscript to Figure 3 in the new version (Fig 3i to l). We also have edited Figure 4 (Fig. 4 g-i) to better show the differences depending on the viral load profiles in vivo.

We also re-analysed the data on the MHC-E dependent NK cell activity (Fig. 5 e-l). In the previous version, we have already shown that IFN-g inhibits SARS-CoV-2 replication in BALF macrophages, but also up-regulates MHC-E on them. This might protect the cells from some NK cell-mediated killing, since that a peptide derived from the Spike protein inhibits MHC-E

dependent NK cell activity. After SARS-CoV-2 infection, some animals showed a modified MHC-E restricted NK cell response to the peptide. We analysed the MHC-E dependent NK cell activity dependent on the level of persisting virus. Here we show now, that NK cells of convalescent animals with the lowest viral load were those able to escape the MHC-E mediated inhibition (Fig 5k, novel graph on the right side).

Allover in the text, we have improved the description of the data and the interpretation of the mechanism of viral persistence with the respective roles of IFN-g, NK cells and NKG2rCD8+ T cells. We have added in the text the limitation that we have not analysed here the role of virus-specific CD8+ T cells, because the scope of the paper was focused on innate immune cells. However, based on the additional analyses and clarifications in the revised manuscript, we hope to provide enough evidence to content the concern of the reviewer regarding the role played by NK cells in persisting SARS-CoV-2 infection and novel key insights into the mechanisms of SARS-CoV-2 persistence in lung-associated tissue.

Reviewer #3

The authors addressed all the concerns I raised, both by performing additional experiments/analyses, and by discussing remaining limitations. Congratulations to a very interesting study.

We thank the reviewer for her/his comment.

We thank all the reviewers for the time dedicated to the manuscript and for the helpful suggestions.

Decision Letter, second revision:
--

7th Aug 2023

Dear Dr. Huot,

Thank you for submitting your revised manuscript "Regulation by IFN- γ and NK cells of SARS-CoV-2 viral persistence in alveolar macrophages of the lung" (NI-A33951B). It has now been seen by the original referees and their comments are below. We are happy to inform you that if you revise your manuscript appropriately according to our editorial requirements, your manuscript should be publishable in Nature Immunology.

I will now pre-edit the current version of your paper. We will also perform detailed checks on your paper and will send you a checklist detailing our editorial and formatting requirements in about two weeks. Please do not upload the final materials and make any revisions until you receive this additional information from us.

In the meantime however, please deposit all omics and code data into public repositories so that the accession codes are readily available to be added in the revised manuscript. We cannot accept the paper without them. In addition, please check that your ORCID (and the ORCID of all the other corresponding authors) is linked to your Nature account, as this frequently causes delays at acceptance. Should you have any query or comments about ORCID, please do not hesitate to contact our editorial assistant at immunology@us.nature.com.

If you had not uploaded a Word file for the current version of the manuscript, we will need one before beginning the editing process; please email that to immunology@us.nature.com at your earliest convenience.

Thank you again for your interest in Nature Immunology. Please do not hesitate to contact me if you have any questions.

Sincerely,

Ioana Visan, Ph.D.
Senior Editor
Nature Immunology

Tel: 212-726-9207
Fax: 212-696-9752
www.nature.com/ni

Reviewer #1 (Remarks to the Author):

This is a new and improved version of this manuscript. All my concerns have been addressed.

Final Decision Letter:

Dear Dr. Huot,

I am delighted to accept your manuscript entitled "SARS-CoV-2 viral persistence in lung alveolar macrophages is controlled by IFN- γ and NK cells" for publication in an upcoming issue of Nature

Immunology.

Over the next few weeks, your paper will be copyedited to ensure that it conforms to Nature Immunology style. Once your paper is typeset, you will receive an email with a link to choose the appropriate publishing options for your paper and our Author Services team will be in touch regarding any additional information that may be required.

Please note that *Nature Immunology* is a Transformative Journal (TJ). Authors may publish their research with us through the traditional subscription access route or make their paper immediately open access through payment of an article-processing charge (APC). Authors will not be required to make a final decision about access to their article until it has been accepted. [Find out more about Transformative Journals](https://www.springernature.com/gp/open-research/transformative-journals).

Authors may need to take specific actions to achieve [compliance with funder and institutional open access mandates](https://www.springernature.com/gp/open-research/funding/policy-compliance-faqs). If your research is supported by a funder that requires immediate open access (e.g. according to [Plan S principles](https://www.springernature.com/gp/open-research/plan-s-compliance)) then you should select the gold OA route, and we will direct you to the compliant route where possible. For authors selecting the subscription publication route, the journal's standard licensing terms will need to be accepted, including [self-archiving policies](https://www.springernature.com/gp/open-research/policies/journal-policies). Those licensing terms will supersede any other terms that the author or any third party may assert apply to any version of the manuscript.

Your paper will be published online soon after we receive your corrections and will appear in print in the next available issue. Content is published online weekly on Mondays and Thursdays, and the embargo is set at 16:00 London time (GMT)/11:00 am US Eastern time (EST) on the day of publication. Now is the time to inform your Public Relations or Press Office about your paper, as they

might be interested in promoting its publication. This will allow them time to prepare an accurate and satisfactory press release. Include your manuscript tracking number (NI-A33951C) and the name of the journal, which they will need when they contact our office.

About one week before your paper is published online, we shall be distributing a press release to news organizations worldwide, which may very well include details of your work. We are happy for your institution or funding agency to prepare its own press release, but it must mention the embargo date and Nature Immunology. Our Press Office will contact you closer to the time of publication, but if you or your Press Office have any enquiries in the meantime, please contact press@nature.com.

Also, if you have any spectacular or outstanding figures or graphics associated with your manuscript - though not necessarily included with your submission - we'd be delighted to consider them as candidates for our cover. Simply send an electronic version (accompanied by a hard copy) to us with a possible cover caption enclosed.

If you have not already done so, we strongly recommend that you upload the step-by-step protocols used in this manuscript to the Protocol Exchange. Protocol Exchange is an open online resource that allows researchers to share their detailed experimental know-how. All uploaded protocols are made freely available, assigned DOIs for ease of citation and fully searchable through nature.com. Protocols can be linked to any publications in which they are used and will be linked to from your article. You can also establish a dedicated page to collect all your lab Protocols. By uploading your Protocols to Protocol Exchange, you are enabling researchers to more readily reproduce or adapt the methodology you use, as well as increasing the visibility of your protocols and papers. Upload your Protocols at www.nature.com/protocolexchange/. Further information can be found at www.nature.com/protocolexchange/about .

Please note that we encourage the authors to self-archive their manuscript (the accepted version before copy editing) in their institutional repository, and in their funders' archives, six months after publication. Nature Portfolio recognizes the efforts of funding bodies to increase access of the research they fund, and strongly encourages authors to participate in such efforts. For information about our editorial policy, including license agreement and author copyright, please visit www.nature.com/ni/about/ed_policies/index.html

An online order form for reprints of your paper is available at <https://www.nature.com/reprints/author-reprints.html>. Please let your coauthors and your institutions' public affairs office know that they are also welcome to order reprints by this

method.

Sincerely,

Ioana Visan, Ph.D.
Senior Editor
Nature Immunology

Tel: 212-726-9207
Fax: 212-696-9752
www.nature.com/ni